

# Evaluation and updates to the oxidized reactive nitrogen trace gas dry deposition parameterization from the GEOS-Chem CTM, including a pathway for ground surface NO₂ hydrolysis

Brian L. Boys[1], Randall V. Martin[2], and Trevor C. VandenBoer[3]

[1]Department of Physics and Atmospheric Science, Dalhousie University, Halifax, NS, Canada
[2]Department of Energy, Environmental, and Chemical Engineering, Washington University in St. Louis, MO, USA
[3]Department of Chemistry, York University, Toronto, ON, Canada

*Correspondence to*: Brian L. Boys (bboys@dal.ca)

**Abstract.** Dry deposition is a major loss pathway for reactive nitrogen species from the atmospheric boundary layer. Represented in chemical transport models (CTMs) as a first-order process, time-varying rate coefficients are parameterized and expressed via species-specific deposition velocities ($V_d(x)$). We evaluate isolated components of the parameterization for $V_d$ in the GEOS-Chem CTM by extracting the trace gas dry deposition algorithm and reimplementing in single-point-mode to enable more direct comparison to field observations. Resistances to surface uptake follow a modified version of the 'big-leaf' Wesely parameterization, which previous studies have shown applies poorly to off-target species such as $NO_2$ under conditions favoring non-stomatal uptake. We evaluate non-stomatal dry deposition of $NO_2$ by comparing to eddy covariance observed nocturnal $V_d(NO_2)$ over Harvard Forest. We eliminate a large low bias (-80 %) in simulated nocturnal $V_d(NO_2)$ by representing $NO_2$ heterogeneous hydrolysis on deposition surfaces, paying attention to chemical flux divergence, soil NO emission, as well as canopy surface area effects. Finally, we evaluate the updated oxidized reactive nitrogen ($NO_y$) dry deposition parameterization for GEOS-Chem by comparing to eddy covariance observed $V_d(NO_y)$ over Harvard Forest, finding a modest nocturnal low bias (-19 %) remains in simulated $V_d(NO_y)$ due to the compensating effects of updates to the calculation of molecular diffusivities (28 % reduction in nocturnal $V_d(NO_y)$) and representation of $NO_2$ heterogenous hydrolysis (25 % increase in nocturnal $V_d(NO_y)$). These developments are applicable to models across scales, having important implications for near-surface $NO_2$ lifetime through a mechanism involving HONO emission.

## 1 Introduction

Chemical species comprising oxidized reactive nitrogen ($NO_y$) form a main component of atmospheric reactive nitrogen ($N_r \equiv NO_y$ + reduced nitrogen species) which together play a central role in atmospheric chemistry by modulating the oxidative capacity of the atmosphere through nitrogen oxides ($NO_x \equiv NO + NO_2$) (Crutzen, 1979), contributing to nitrogen loading of natural ecosystems (Clark et al., 2018), and influencing air quality (Fields, 2004). Accurate knowledge of sources and sinks of $N_r$ is vital for understanding and modeling atmospheric chemistry, including the sensitivity of air quality to changes in



anthropogenic emissions. Dry deposition of $N_r$ from the atmospheric boundary layer is an important removal process, typically contributing between one-third to two-thirds of total (wet + dry) deposition (Flechard et al., 2011; Hanson & Linderg, 1991; Munger et al., 1998; Sparks et al., 2008; Walker et al., 2020), but questions remain about its representation in chemical transport models (CTMs).

The atmosphere–surface exchange of $N_r$ may be measured directly via micrometeorological techniques (Businger, 1985; Walker et al., 2020) or under more controlled conditions via enclosure techniques (Breuninger et al., 2012; Hanson and Linderg, 1991). Direct measurements of above-canopy air–surface exchange of $N_r$, including via the eddy covariance technique, are technically complex and resource intensive, resulting in a scarcity of flux observations across representative land types and seasons (Walker et al., 2020). Therefore, studies of above-canopy dry deposition tend to be intensive in nature and are typically designed to characterize exchange processes rather than to monitor long-term deposition patterns. Dry

deposition budgets thus fall to the realm of inferential methods, where deposition fluxes $F_x$ are inferred from parameterizations of above-canopy deposition velocity $V_d$—a first-order rate coefficient for heterogenous surface reaction/uptake for a specific gas $x$ to a specific bulk surface/land type from a specified height:

$$F_x = -V_d(x)\,[x]\,, \tag{1}$$

By convention, downward fluxes toward the surface are negative values represented by positive deposition velocities. $N_r$

component concentrations $[x]$ from which dry deposition budgets may be inferred have been obtained from: (i) surface networks such U.S. CASTNET (Clarke et al., 1997) and Canadian CAPMoN (Zhang et al., 2009), (ii) chemical transport models (Dennis et al., 2013; Zhang et al., 2012; Zhang et al., 2018), and (iii) satellite observations (Geddes & Martin, 2017; Kharol et al., 2018; Nowlan et al., 2014).

Deposition velocity represents a bulk quantity with contributions from complex processes including turbulent and

molecular diffusion in air, meteorological influence on the physical, chemical, and biological state of surfaces, and species-specific interfacial chemistry. The most common parameterization of $V_d$ in large-scale CTMs considers the deposition pathway as a series of three resistances (Baldocchi et al., 1987; Wesely and Hicks, 1977):

$$V_d(x) = \frac{1}{R_a(z) + R_b(x) + R_c(x)}\,, \tag{2}$$

where for bulk-canopy $V_d$ above a projected ground area, $R_a(z)$ is the aerodynamic resistance to turbulent transport from a

specified height $z$ and is common for all species, $R_b(x)$ is the species-specific quasi-laminar boundary layer resistance to transport through the thin nonturbulent layer in direct contact with surfaces, and $R_c(x)$ is the bulk-canopy surface resistance for a specific land type. Expressions for $R_a$ and $R_b$ can be obtained from micrometeorological flux–gradient relationships (Garratt, 1992) and vary as a function of surface roughness, wind speed, diabatic stability, and molecular diffusivity in air. For highly soluble species such as $HNO_3$ and $H_2O_2$, contributions from $R_c$ are nominally small with resulting deposition varying between

$R_a$- and $R_b$-limited depending on the state of turbulence (Nguyen et al., 2015). For species with low aqueous solubility or limited interfacial reactivity, $R_c$ is the limiting term, except under very stable conditions (Toyota et al., 2016). Given the





complexity and variability of canopy types and species-specific surface reactivities, $R_c$ is difficult to treat theoretically, with parameterizations relying heavily on empirical formulations.

The most common parameterization of $R_c$ used by large scale atmospheric models, including the widely utilized WRF-Chem and GEOS-Chem CTMs, is the Wesely 1989 algorithm (Wesely, 1989; hereafter referred to as W89), or modifications thereof (Hardacre et al., 2015). In this scheme, the bulk-canopy is treated as a single uniform surface or 'big leaf' with stomatal and various non-stomatal deposition pathways acting in parallel. Trace gas specific component surface resistances are calculated following basic similarity relations, including solubility relative to $SO_2$ and oxidative potential relative to $O_3$. Zhang et al. (2003) present a parameterization of $R_c(x)$ for use in air quality models, including at the global scale, employing similarity arguments to $SO_2$ and $O_3$ as done in W89, with several updates to the scheme including online computation of within canopy aerodynamic resistance, influence of leaf water vapour pressure deficit and water stress on stomatal resistance, and updated parameterizations of non-stomatal surface resistances for $O_3$ (Zhang et al., 2002) and $SO_2$ (Zhang et al., 2003). In these chemical deposition updates, Zhang et al. (2003) note that application of the algorithm (hereafter referred to Z03) to compounds for which little to no deposition flux observations exist will continue to be a source of significant uncertainty and a call was made to increase efforts to study species-specific fluxes across representative land types and seasons.

Bulk-canopy surface resistances deviating from similarity to $SO_2$ and $O_3$ have been observed for $NO_2$ (Eugster & Hesterberg, 1996; Horii, 2002; Stella et al., 2013), PAN (Shepson et al., 1992; Sun et al., 2016; Turnipseed et al., 2006), and many other species (Nguyen et al., 2015). Wu et al. (2012) compare observed $V_d(PAN)$ over a coniferous forest to deposition velocities parameterized according to both the W89 (WRF-Chem) and Z03 (NOAH) schemes and find underestimates greater than a factor of 2, motivating an effort to fit non-stomatal $R_c(PAN)$ directly from above-canopy flux observations. Flechard et al. (2011) compare dry deposition fluxes of reactive nitrogen species $NH_3$, $NO_2$, $HNO_3$, HONO, particulate ammonium ($pNH_4$) and nitrate ($pNO_3$) across an inferential network of 55 sites throughout Europe using four existing dry deposition routines and note differences between models (up to a factor of 2 to 3) are often greater than differences between sites, calling for more long-term direct $N_r$ flux measurements with which to validate dry deposition algorithms.

A main result of Horii et al. (2004) in their analysis of an extensive eddy covariance flux dataset of $NO_2$ over a Northeastern U.S. mixed forest (Harvard Forest) from April–November was that a persistent deposition process was active at night, yielding $NO_2$ deposition velocities on average of $\sim 0.2$ cm s$^{-1}$, with values up to 0.5 cm s$^{-1}$ noted under high $NO_2$ loads of $\sim 30$ ppb. This observation is contrary to the widely used W89 parameterization which does not allow significant surface uptake of $NO_2$ at night when leaf stomata are assumed closed or during vegetatively dormant seasons. Geddes et al. (2014) monitored eddy covariance fluxes of NO, $NO_2$, and $NO_y$ above midlatitude ($\sim 45°$ N) summertime mixed hardwood forests in Ontario (Canada) and Michigan (U.S.), finding on average $NO_x$ fluxes indistinguishable from zero for these relatively low $NO_y$ environments (< 2 ppb on average). However, infrequent nocturnal events with high $NO_x/NO_y$ ratios and large downward $NO_y$ fluxes could be interpreted as yielding $NO_2$ deposition velocities similar to the average values of Horii et al. (2004). Geddes et al. (2014) were careful to note that above-canopy fluxes of $NO_x$ are influenced not only by deposition processes but also by within canopy emissions and chemistry, resulting in above-canopy fluxes of $NO_x$ that are confounded by a combination



of counteracting mechanisms which render flux observations difficult to interpret. Horii et al. (2004) considered below-sensor chemical flux divergence of $NO_2$ due to formation and subsequent hydrolysis of $N_2O_5$, showing that a maximum rate of loss is insufficient to account for observed downward nocturnal $NO_2$ flux and propose a non-stomatal hydrolysis pathway for uptake of $NO_2$ on ground and canopy surfaces—a reaction which has been suspected in the field to be of atmospheric relevance for some time (Harrison & Kitto, 1994; Harrison et al., 1996).


The hydrolysis of $NO_2$ on hydrated surfaces is a well-known heterogenous reaction yielding adsorbed $HNO_3$ and evolved nitrous acid (HONO):

$$2\,NO_{2\,(g)} + H_2O_{(ads)} \xrightarrow{\text{surface}} HONO_{(g)} + HNO_{3\,(ads)}\,, \tag{R1}$$

Despite the stoichiometry of reaction R1, first-order kinetics in $NO_2$ have generally been observed, with a rate having
dependence on surface area density (as expected for collision-limited heterogeneous catalysis), surface water content, and other surface chemical properties (Finlayson-Pitts, 2009; Finlayson-Pitts et al., 2003; Lammel, 1999; Spataro and Ianniello, 2014). In addition to hydrated ground (Kurtenbach et al., 2001; Lammel, 1999; Ren et al., 2020; VandenBoer et al., 2013) and aerosol (Bröske et al., 2003; Burkholder et al., 2015; Crowley et al., 2010; Tan et al., 2016) surfaces, reaction R1 has been implicated on the sea surface (Wojtal et al., 2011; Yang et al., 2021; Zha et al., 2014), on snow and ice surfaces (Beine et al.,
2001; Kim and Kang, 2010), as well as on indoor surfaces (Collins et al., 2018; Febo and Perrino, 1991; Spicer et al., 1993). Spicer et al. (1993) and Collins et al. (2018) both found an indoor lifetime of $NO_2$ to reactive loss (HONO producing) on residential interior surfaces on the order of one hour in well mixed air—lower than typical ambient $NO_2$ chemical lifetimes on the order of hours in regional (Kenagy et al., 2018; Shah et al., 2020) and urban (Laughner and Cohen, 2019) outflows and remote forest environments (Browne and Cohen, 2012). Reaction R1 may be an especially important surface removal process
during summertime nights or winter months when $NO_2$ is longer lived with lifetimes on the order of 10 h to more than a day (Browne and Cohen, 2012; Kenagy et al., 2018; Martin et al., 2003). Reaction R1 has also been implicated in the uptake of $NO_2$ through leaf stomata, where it may be an important contributor to $NO_2$ deposition within the moist and high surface area substomatal cavities (apoplast) of leaves (Ammann et al., 1995).

Despite the evidence for reaction R1 proceeding on outdoor and indoor surfaces, regional and global CTMs have yet,
to our knowledge, to update dry deposition parameterizations of $NO_2$ to include this effect, potentially underestimating $V_d(NO_2)$ at night and throughout vegetatively senescent periods when stomatal uptake would be weak or absent. In this study, we extract the trace gas dry deposition parameterization from the GEOS-Chem global CTM and reimplement to run in single-point-mode to facilitate evaluation of an updated parameterization that includes the effect of reaction R1 on simulated $V_d(NO_2)$ and $V_d(NO_y)$. We compare to above-canopy observations of $V_d(NO_2)$ and $V_d(NO_y)$ inferred from an extensive publicly available dataset of
$NO_2$ and $NO_y$ eddy covariance fluxes and speciated $NO_y$ concentration measurements over Harvard Forest, Massachusetts, U.S. (Munger and Wofsy, 2023), paying attention to soil NO emission and canopy surface area effects. Prior to updating simulated $R_c(NO_2)$ to include reaction R1 on deposition surfaces, we evaluate the parameterization of $R_a$ and $R_b$ from GEOS-Chem by comparing to daytime deposition velocities of rapidly depositing species inferred by the method of eddy covariance



over a Southern U.S. temperate forest (Nguyen et al., 2015). Specifically, we comment on the effects that site-specific

roughness length, reference height, and the roughness sublayer have on the simulation of daytime $R_a$, followed by correction

of a positive bias in calculated molecular diffusivities which greatly improves the simulation of daytime $V_d(HNO_3)$ via a large

relative increase in $R_b$.

## 2 Reference Model and Measurements

### 2.1 Trace gas dry deposition parameterization from the GEOS-Chem model

To facilitate site-specific comparisons to measured deposition velocities, we extract the trace gas dry deposition source code

and input parameters from GEOS-Chem v10-01 (www.geos-chem.org) and implement in single-point-mode with the option

to use on-site meteorology and phenological characterizations (i.e., LAI, canopy height, and land type classification). The dry

deposition flux of trace gases in GEOS-Chem proceeds in grid cells in contact with the ground following an inferential

technique (Eq. (1)), with species-specific deposition velocities $V_d(x)$ computed following the standard resistance-in-series

approach (Eq. (2)).

Aerodynamic resistance is formulated to represent the resistance to turbulent transport of scalars within the surface

layer from a reference height $z$ (i.e., a measurement height or model grid box center) down to the roughness length $z_o$ of the

surface—the height above the zero plane displacement $d$ where the logarithmic wind profile is assumed to extrapolate to zero

(Garratt, 1992; Kaimal and Finnigan, 1994; Toyota et al., 2016; Wesely and Hicks, 1977):

$$R_a(z) = \frac{1}{ku_*}\left[\ln\left(\frac{z-d}{z_o}\right) - \Psi_h\left(\frac{z-d}{L}\right) + \Psi_h\left(\frac{z_o}{L}\right)\right],\qquad(3)$$

where $k$ is the von Karman constant (0.4 in GEOS-Chem), $u_*$ the friction velocity—a surface layer velocity scale which

characterizes surface momentum flux, and $\Psi_h$ an integrated Monin-Obukhov (M-O) stability-correction factor for sensible

heat (Section S1 in the supplement)—an empirical function of the dimensionless ratio $(z - d) / L$ where $L$ is the M–O length

(Monin and Obukhov, 1954). Both $z_o$ and $d$ are fit parameters to the logarithmic wind profile under neutral stability (Monin

and Obukhov, 1954). Empirical values typical of natural vegetated surfaces are: $z_o \sim 1/10$ canopy height ($h_c$) and $d \sim 2/3\ h_c$

(Garratt, 1992; Oke, 1987). At heights well above the surface ($z > 10\ h_c$), $d$ may be ignored in the calculation of $R_a$ (Garratt,

1992), as is done in GEOS-Chem since dry deposition is referenced from surface grid box centers ($z \sim 60$ m AGL). Equation

(3) applies equally to all trace gas and aerosol species and assumes equivalency in the turbulent transfer of momentum and

scalers under neutral conditions from $z$ to $z_o$. It is noted that $R_a$ according to Eq. (3) assumes a 'no-slip' boundary condition,

that is, $u(z_o) = 0$ m s$^{-1}$—the implications of which are discussed in Section 3.1.

Across the distance $z_o$, molecular diffusion becomes an important factor governing near-surface trace gas flux. The

species-specific quasi-laminar boundary layer resistance accounts for the transfer of gases from $z_o$ to the deposition surface,

and is estimated using the semi-empirical formulation of Wesely and Hicks (1977):



$$R_b(x) = \frac{2}{ku_*}\left(\frac{\kappa}{D_x}\right)^{2/3},$$  (4)

where $\kappa$ is the thermal diffusivity of air and $D_x$ the molecular diffusivity of the depositing trace gas $x$. Developments made herein to the calculation of $D_x$ from GEOS-Chem are discussed in Section 3.2.

The resistance to surface uptake of trace gases in GEOS-Chem is parameterized according to a modified 'big leaf' algorithm based on the W89 scheme, as is currently the case for the majority of global CTMs (Hardacre et al., 2015). Species-specific bulk-canopy surface resistance $R_c(x)$ is computed as multiple deposition pathways acting in parallel, including to: (i)

upper canopy leaf interiors via stomatal $r_s$ and mesophyll $r_m$ resistances, (ii) upper canopy leaf cuticles $r_{lu}$, (iii) lower canopy elements $r_{dc} + r_{cl}$, and (iv) ground surface elements $r_{ac} + r_{gc}$:

$$R_c(x) = \left[1/(r_s + r_m) + 1/r_{lu} + 1/(r_{dc} + r_{cl}) + 1/(r_{ac} + r_{gc})\right]^{-1},$$  (5)

The W89 algorithm was originally developed over the U.S. and southern Canada for use on 11 land types, with component resistances varying across 5 seasonal categories (summer, autumn, late autumn, winter, spring). Application to a variety of

trace gases was made possible by $r_s$ dependence on molecular diffusivity and $r_m, r_{lu}, r_{cl}$, and $r_{gc}$ dependence on (i) aqueous solubility at neutral pH via effective Henry's solubility ($H^*$) and (ii) oxidative capacity via an estimated reactivity factor ($f_o$) categorized as unreactive ($f_o = 0$), slightly reactive ($f_o = 0.1$), or as reactive as $O_3$ ($f_o = 1$). Categorized $f_o$ values are based on electron activities and rate-of-reaction with aqueous S(IV) compounds (Wesely, 1989). In-canopy aerodynamic resistance to turbulent transport to the lower canopy and ground surface is represented by land type dependent fixed values $r_{dc}$ and $r_{ac}$,

respectively. Implementation of the W89 algorithm into GEOS-Chem included modifications for application to the global scale (Wang et al., 1998). Detailed descriptions of these modifications have been included in recent work evaluating the dry deposition of $O_3$ in GEOS-Chem (Silva and Heald, 2018; Wong et al., 2019) and can be found online at http://wiki.seas.harvard.edu/geos-chem/index.php/Dry_deposition (last accessed on 01/12/2023). Following the recommendations of Shah et al. (2018), we limit the cold temperature exponential increase in the non-stomatal components of

$R_c$ to a factor of 2 and impose a nominally small $R_c(HNO_3) = 1$ s m$^{-1}$.

Meteorological inputs to the parameterization of $V_d$ in GEOS-Chem are provided from assimilated meteorological fields from NASA's Global Modeling and Assimilation Office (GMAO). Daily LAI values are interpolated from a gridded MODIS-derived monthly LAI product (Myneni et al., 2002). As previously mentioned, we have implemented the option to use on-site meteorology and phenological characterizations to drive the GEOS-Chem dry deposition scheme reimplemented

herein to run in single-point-mode.

**2.2 Above-canopy deposition velocities inferred from eddy covariance measurements**

We evaluate the extracted trace gas dry deposition scheme from GEOS-Chem against eddy covariance observed deposition velocities over two temperate forests in the U.S. First, we compare to deposition velocities from Nguyen et al. (2015) for species found to dry deposit with minimal surface resistance, namely, $H_2O_2$, hydroxy methylhydroperoxide (HMHP), and



HNO$_3$, over Talladega National Forest in June. Second, for an in-depth evaluation of simulated $V_d(NO_2)$ and $V_d(NO_y)$, we use a publicly available long-term hourly dataset of eddy covariance flux observations of NO$_2$ and NO$_y$ from the Harvard Forest Environmental Monitoring Site, supported with ancillary measurements including NO$_y$ component concentrations and meteorological and phenological variables.

### 2.2.1 Talladega National Forest: H$_2$O$_2$, HMHP, and HNO$_3$

Nguyen et al. (2015) present a novel dataset containing eddy covariance observed deposition velocities of 16 trace gases, including species with negligible surface resistance: H$_2$O$_2$, HMHP, and HNO$_3$. Being able to neglect the complexities of a surface resistance scheme allows for a more direct evaluation of $R_a$ and $R_b$ components of the resistance-in-series pathway used in the parameterization of $V_d$ in GEOS-Chem. Observations were taken at the Centreville (CTR) Southeastern Aerosol Research and Characterization Study (SEARCH) site (32.90289° N, 87.24968° W) near Brant, Alabama, U.S. in June 2013.

The CTR site is situated in a grassy clearing in the Talladega National Forest with large forest fetch to the N, W, and E. The mixed forest consists of coniferous and deciduous tree species, with a mean canopy height of ~10 m and LAI of 4.7 m$^2$ m$^{-2}$. Eddy covariance flux observations were measured at 22 m AGL from a walk-up tower, with sonic anemometer (8 Hz) and inlet to the time-of-flight (TOF) chemical ionization mass spectrometer (CIMS) (10 Hz) facing north so as to capture eddies originating over forest fetch. The analysis of Nguyen et al. (2015) includes daytime mean deposition velocities averaged

between 10–15 Local Solar Time (LST) across five ideal days in June 2013 (6$^{th}$, 15$^{th}$, 20$^{th}$, 23$^{rd}$, and 27$^{th}$) when winds had exclusively forest fetch. To compare with the reported daytime deposition velocities of H$_2$O$_2$ (5.2 ± 1.1 cm s$^{-1}$), HMHP (4.1 ± 1.1 cm s$^{-1}$), and HNO$_3$ (3.8 ± 1.3 cm s$^{-1}$), we average $R_a$ and R$_b$ components of the offline dry deposition algorithm, applied at the location of the CTR site, between 10–15 LST on the aforementioned days. Meteorological inputs required to compute $R_a$ and $R_b$ components of the algorithm ($u_*$, $T$, $P$, and sensible heat flux) were obtained from NASA's Goddard Earth Observing

System (GEOS) Forward Processed (FP) assimilated meteorological fields (Lucchesi, 2013) at the native horizontal resolution of 1/4° x 5/16°, which Nguyen et al. (2015) note are in excellent agreement with values measured at the CTR site during the measurement period.

### 2.2.2 Harvard Forest: NO$_2$ and NO$_y$

The utility of the Harvard Forest Environmental Monitoring Site (HFEMS) for evaluating parameterizations of atmosphere–

surface exchange stems from the extensive datasets of meteorological, phenological, and trace gas observations spanning many months to years at high temporal (hourly) resolution. The HFEMS is located in central Massachusetts, U.S. (42.54° N, 72.18° W; 340 m ASL) and situated in a mature mixed deciduous forest ($h_c$ ~ 20 m) with a summertime LAI of 4.3 and deciduous LAI of 3.4 (Fig. S4 in the supplement). Local pollution sources include a secondary paved road 1.5 km to the west, a two-lane expressway ~ 5 km to the north, and a small town more than 10 km to the northwest. Due to prevailing westerly winds,

emissions from Boston (100 km to the east) rarely influence the site. Cool, dry, and unpolluted air from the northwest and




warm, moist, anthropogenically influenced air from the southwest are the predominant influences at this site (Horii et al., 2005).

Munger et al. (1996) describe the methodology of long-term total nitrogen oxide ($NO_y$) concentration measurements for eddy covariance flux computation, as well as other details of the HFEMS. Briefly, $NO_y$ concentrations at 8 Hz were made
by reducing $NO_y$ to NO on a well-aged hot gold catalyst with $H_2$, followed by detection of chemiluminescence from titration of resulting NO with $O_3$. The reducing catalyst was positioned close to the inlet at a height of 29 m on the 30 m walk-up tower. Concentration measurements of PAN by capillary-column gas chromatograph with electron capture detection was added to the 30 m walk-up tower in April 2000 (Horii et al., 2005). To an auxiliary 23 m scaffolding tower located ~ 100 m to the southeast of the main tower, a Tunable Diode Laser Absorption Spectrometer (TDLAS) was configured to measure eddy
covariance fluxes (1 Hz) of $NO_2$ and concentrations of $HNO_3$ from April through November 2000 (Horii et al., 2004). Due to inlet wall interactions of $HNO_3$ with a characteristic time constant of ~ 10 minutes, high frequency concentration information required for eddy covariance flux computation was not possible; however, it was found that the hourly mean concentration was not compromised, as the fluorinated silane-coated fused silica quartz inlet walls were not a permanent sink of $HNO_3$ which was near completely transmitted to the measurement cell after sufficient equilibration time (Horii et al., 2005). Although the
measurement height of $NO_y$, NO, $NO_2$, and PAN on the main tower (29 m) did not match that of $HNO_3$ and $NO_2$ on the auxiliary tower (22 m), Horii et al. (2005) found the measurement heights of the two towers to be in the same flux regime by congruence of heat fluxes and noted as well the coherence in coincident trace gas data on the hourly timescale.

Trace gas data from the HFEMS used in this study, specifically hourly eddy covariance fluxes of $NO_y$ and $NO_2$, and hourly concentrations of $NO_y$, NO, $NO_2$, PAN, and $HNO_3$ are publicly available from the Harvard Forest Data Archive
(Munger and Wofsy, 2004, 2023). Exchange velocities ($V_{ex}$) are computed herein by normalizing reported hourly $NO_y$ and $NO_2$ eddy covariance fluxes by respective ambient hourly concentrations. Equating $V_{ex}$ to $V_d$ assumes that the observed flux is due to surface deposition only. Processes causing deviation from this assumption are discussed in later sections and include surface emission of NO, chemical flux divergence of $NO_2$, and a potential non-zero canopy accumulation rate of $NO_y$. Eddy covariance fluxes have reduced error under conditions where turbulence is well developed (Baldocchi, 2003; Cherin et al.,
2015; Goulden et al., 1996; Nguyen et al., 2015). Turbulent threshold $u_*$ values in the range 0.15–0.35 m s$^{-1}$ (median 0.23 m s$^{-1}$) have been found to be representative of multiple sites across many years (Cherin et al., 2015). Herein, periods of low surface layer turbulence ($u_* < 0.2$ m s$^{-1}$) have been omitted from analysis, resulting in ~ 25 % of hourly values of nocturnal $V_d(NO_2)$ and 18 % of hourly values of $V_d(NO_y)$ being removed from the HFEMS dataset. Outliers in the remaining hourly $V_d(NO_2)$ and $V_d(NO_y)$ timeseries were identified via the method of median absolute deviation (MAD) (Leys et al., 2013), where
hourly values outside of the median ± 3x MAD were removed from calculations of subsequent means; ~ 20 % of hourly nocturnal $V_d(NO_2)$ and 10 % of hourly $V_d(NO_y)$ were removed from the $u_*$-filtered dataset. Overall, 60 % of the nocturnal $V_d(NO_2)$ and 74 % of the $V_d(NO_y)$ hourly timeseries were retained for analysis after application of turbulence and outlier filters.

Meteorological input variables required in the parameterization of $V_d$ were taken from the HFEMS data archive; specifically, $P$, $T$, $RH$, $u_*$, and sensible heat flux (Munger and Wofsy, 2024) and incoming solar radiation (Fitzjarrald and



Sakai, 2023) were available at hourly temporal resolution throughout the study period. Cloud fraction was the only required meteorological variable not available from the HFEMS data archive and was instead taken from NASA's Modern-Era Retrospective analysis for Research and Applications version 2 (MERRA-2) assimilated meteorological fields (Gelaro et al., 2017). Figure S5 in the supplement depicts comparisons of hourly observations of $u_*$, sensible heat flux, downward shortwave radiation, $T$, $P$, and $RH$ made over Harvard Forest to coincident values from MERRA-2 assimilated meteorology. Excellent

agreement (normalized mean bias NMB < 1 % and $R^2$ > 0.93) is noted for $T$ and $P$; $RH$ and $u_*$ have small biases of -6 % and 5 %, respectively, with an $R^2$ of 0.60 and 0.69, respectively. Sensible heat flux and downward shortwave radiation are each biased high by 16 %, with an $R^2$ of 0.68 and 0.87, respectively. Canopy-specific inputs to the parameterization of $V_d$ include $z_o$, $d$, LAI, and deposition land type. Land type was set to deciduous with values for $z_o$ and $d$ being estimated as 1/10th and 2/3rd of canopy height, respectively—values representative of many vegetative surfaces including forests (Garratt, 1992; Oke, 1987).

We estimate daily LAI values from a spline-fit to daily Plant Area Index (PAI) measurements from the HFEMS over April–December for years 1998–2015 (Matthes et al., 2024), corrected for the reported stem and twig area index (STAI) of 0.9 $m^2$ $m^{-2}$ noted for this canopy (Horii et al., 2004). Estimated climatological daily LAI values range from ~ 0.9 $m^2$ $m^{-2}$ in winter to 4.3 $m^2$ $m^{-2}$ in summer, in good agreement with MODIS-derived LAI at the location of Harvard Forest (Fig. S4).

**2.3 Measured diffusion coefficients of atmospherically relevant molecules**

A main result of Nguyen et al. (2015) was the importance of molecular diffusion in atmosphere–surface exchange of rapidly depositing compounds, where it was shown that maximum daytime dry deposition velocities scale with the inverse square root of molecular mass ($M^{-1/2}$), as do gas phase diffusion coefficients (Poling and Prausnitz, 2004). To evaluate the calculation of molecular diffusivities used in the parameterization of dry deposition velocities in GEOS-Chem, we conducted a literature search to compile a list of measured diffusion coefficients of atmospherically relevant molecules for 23 inorganic and 17

organic species (Table S1). Diffusion coefficients ($D$) measured in either air or $N_2$ near STP were corrected to STP following Langenberg et al. (2020):

$$D = D_o \left(\frac{P_o}{P}\right) \left(\frac{T}{T_o}\right)^b , \qquad (6)$$

where we set the temperature power dependence $b$ = 1.75 following Fuller's method, a semi-empirical technique for the estimation of binary gas-phase diffusion coefficients (Fuller et al., 1966), discussed further in Section 3.2.

**2.4 Measurements of surface-specific deposition velocities for NO₂**

Surface-specific NO₂ uptake coefficients ($\gamma_{NO_2}$) to both foliar and non-foliar forest elements facilitate bottom-up estimates of bulk-canopy $R_c(NO_2)$ and resulting $V_d(NO_2)$ to forest environments when corresponding surface area scale factors (i.e., DLAI, CLAI, and STAI) and meteorological data are available. From literature values of surface-specific deposition velocities $v_d^{surf}$, we infer NO₂ uptake coefficients $\gamma_{NO_2}$ to both non-foliar and foliar materials:





$\gamma_{NO_2} = \dfrac{4\, v_d^{surf}}{\overline{v}_t}$,            (7)

where $\overline{v}_t$ is the mean thermal speed of $NO_2$. Table S2 contains literature values of $v_d^{surf}$ with associated surface area assumptions, experimental temperatures, and relative humidities which were used to infer values of $\gamma_{NO_2}$ to foliar surfaces of deciduous and coniferous species under nocturnal/dark conditions, non-foliar forest materials (bark and forest floor), snow, and fabricated materials. With the exception of deposition to snow, literature values of $v_d^{surf}$ are from chamber studies where

mechanically mixed chamber air enables the direct estimation of $\gamma_{NO_2}$ through Eq. (7), i.e., turbulent ($R_a$) and quasi-laminar ($R_b$) resistances may be neglected for species with slow surface uptake (such as $NO_2$) since $R_c \gg R_a + R_b$. Values of leaf-level $v_d^{surf}$ for both deciduous and coniferous species were averaged across periods of minimum stomatal conductance resulting from the absence of photosynthetically active radiation (PAR) or the influence of abscisic acid (ABA) and are interpreted herein for the purpose of computing resulting $\gamma_{NO_2}$ as non-stomatal. Surface areas used for flux normalization are reported

when available. Care must be taken when comparing surface-specific $V_d{}^{surf}$ and $\gamma_{NO_2}$, as various surface area indices are used (i.e., planar, geometric, LAI, and total leaf area). Some studies report $V_d{}^{surf}$ to coniferous species normalized to total leaf area (Breuninger et al., 2013; Hanson et al., 1989) as stomata are distributed across the whole needle surface (amphistomatic), while others normalize to projected LAI as is routinely done for deciduous leaves which generally have stomata on the lower (abaxial) leaf surface; failing to recognize this difference would result in a misrepresentation of $V_d{}^{surf}$ and inferred $\gamma_{NO_2}$ by a factor of ~

2.7 for coniferous species (Riederer et al., 1988).

       Table 1 summarizes values of $\gamma_{NO_2}$, including those used in Section 3.3.4 to compute nocturnal bottom-up estimates of bulk-canopy $R_c(NO_2)$ and resulting $V_d(NO_2)$ over Harvard Forest. Also included are associated surface area scale factors over which measured uptake was normalized and relative humidities over which measurements were made. We suggest that $NO_2$ uptake to the surfaces listed in Table 1 may result from heterogenous hydrolysis of $NO_2$ following reaction R1, with

variability between surfaces primarily a result of differences in microscopic surface area supporting adsorbed water. Some of the studies measuring foliar uptake of $NO_2$ under conditions where stomatal aperture should be at a minimum conclude that uptake could occur to the interior of leaves via partially open stomata rather than non-stomatally to the exterior leaf surfaces (Chaparro-Suarez et al., 2011; Delaria et al., 2020; Rondón et al., 1993). Our assumption of nocturnal stomatal closure with deposition of $NO_2$ to the exterior of leaves is discussed in Section 4. To help contextualize values of $\gamma_{NO_2}$, Section S2 in the

supplement provides a brief literature review of uptake coefficients for $NO_2$ to hydrated surfaces.



**Table 1:** Surface-specific $NO_2$ uptake coefficients $\gamma_{NO_2}$ inferred from literature values of surface-specific deposition velocities[a] following Eq. (7). Also included are corresponding surface area scale factors over which $\gamma_{NO_2}$ is to be applied and relative humidities over which deposition measurements were made.

| Material | $\gamma_{NO_2}$ | Surface Area[b] | RH | Ref. [f] |
|---|---|---|---|---|
| | (unitless) | | [%] | |
| **Non-Foliar surfaces** | | | | |
| distilled water | $2.3 \times 10^{-6}$ | total (planar) | N/A | 1 |
| wood board (untreated, | $7.6 \times 10^{-7}$ | geometric | 70 | 2 |
| hard, fine) | $1.6 \times 10^{-6}$ | | 90 | |
| plywood (untreated) | $1.4 \times 10^{-6}$ | geometric | 50 | 2 |
| tree bark (dry) [c] | $5.0 \times 10^{-6}$ | geometric | unknown | 1 |
| tree bark (wet) [c] | $1.0 \times 10^{-5}$ | geometric | N/A | |
| forest floor[c] | $4.3 \times 10^{-5}$ | planar | ~60 +/- 20 | 3 |
| snow[c] | $1.6 \times 10^{-5}$ | planar | N/A | 4 |
| **Foliar surfaces**[d] | | | | |
| deciduous leaves [c, e] | $1.7 \times 10^{-6}$ | LAI | 50 to < 90 | 1, 6–8 |
| coniferous leaves | $4.6 \times 10^{-6}$ | LAI | 50 to < 90 | 1,3,5,6,8,9 |
| coniferous leaves [c, e] | $1.7 \times 10^{-6}$ | total leaf area | 50 to < 90 | 1,3,5,6,8,9 |

[a] Surface-specific deposition velocities were taken from chamber studies, with the exception of uptake to snow which was measured via the eddy covariance technique. Table S2 in the supplement contains study specific details.

[b] Surface area used to normalize material-specific deposition fluxes in the computation of material-specific $v_d^{surf}$.

[c] Values used in Section 3.3.4 to compute bottom-up estimates of nocturnal bulk-canopy $V_d(NO_2)$ over Harvard Forest.

[d] Foliar uptake was measured under conditions of minimal stomatal aperture, i.e., dark conditions. We assume this uptake to be non-stomatal (Section 4).

[e] Multi-study mean value computed herein (Table S2).

[f] References for material-specific $v_d^{surf}(NO_2)$

(1) Hanson et al. (1989)
(2) Grøntoft and Raychaudhuri (2004)
(3) Rondón et al. (1993)
(4) Stocker et al. (1995)
(5) Wang et al. (2020)
(6) Delaria et al. (2020)
(7) Delaria et al. (2018)
(8) Chaparro-Suarez et al. (2011)
(9) Breuninger et al. (2013)

## 3 Measurement–model comparisons and updates

Table 2 summarizes modifications made herein to the offline trace gas dry deposition parameterization from GEOS-Chem, discussed in-turn throughout this section. Briefly, parameterization P1 is equivalent to the dry deposition scheme in GEOS-Chem, which references deposition from grid box centers (GBC) of the lowest model level (~ 60 m). Serial modifications to P1 include changes to the height in which dry deposition is referenced (P2), formulations to the calculation of aerodynamic resistance (P3, P4) and molecular diffusivities (P5), updating non-stomatal surface resistance for $NO_2$ to include heterogeneous



hydrolysis on deposition surfaces (P6, P7), and implementation of empirical updates to the non-stomatal uptake of PAN (P8).

We evaluate parameterizations P1–P5 by comparing to measured dry deposition velocities from Nguyen et al. (2015), where it was noted that above-canopy deposition velocities for $H_2O_2$, HMHP, and $HNO_3$ corresponded to computed theoretical maximums, thus enabling a more direct evaluation of the deposition pathway consisting of resistances $R_a$ and $R_b$, as discussed in Sections 3.1 and 3.2, respectively. Parameterizations P6–P7 are evaluated by comparing to both above-canopy nocturnal $V_d(NO_2)$ observed at the HFEMS (Section 3.3.3) and bottom-up estimates of nocturnal $V_d(NO_2)$ for Harvard Forest from

literature values of surface-specific deposition velocities $V_d^{surf}(NO_2)$ (Section 3.3.4). Parameterization P8 is evaluated in Section 3.4 in the context of effects on simulated $V_d(NO_y)$, including comparison to above-canopy diel $V_d(NO_y)$ observed at the HFEMS.

**Table 2:** Modifications to the offline dry deposition parameterization tested in this study. Parameterization P1 is equivalent to the trace gas dry deposition scheme in GEOS-Chem (GC). Modifications to P1 include changes to reference height $z_{ref}$ (P2), formulation of aerodynamic
resistance $R_a$ (P3–P4), molecular diffusivity $D$ (P5), and non-stomatal surface resistances ($R_c$) for NO2 (P6–P7) and PAN (P8).

| Param. | $z_{ref}$[a] | Aerodynamic Res. $R_a$ | Diffusivity $D$ | non-stomatal $R_c(NO_2)$ | non-stomatal $R_c(PAN)$ |
|---|---|---|---|---|---|
| P1 | $z_{GBC}$ | base GC (Eq. 3) | base GC (Chapman–Enskog theory with constant mfp[b] | base GC (modified W89) | base GC (modified W89) |
| P2 | $z_{TNF}$ | | | | |
| P3 | or | RSL, $u(z_o) > 0$ m s$^{-1}$ | | | |
| P4 | $z_{HFEMS}$ | RSL, $u(z_o) = 0$ m s$^{-1}$ | | | |
| P5 | | | measured & Fuller's method (Eq. 6 & 8) | | |
| P6 | | RSL, $u(z_o) > 0$ m s$^{-1}$ | | $r_{hyd}$ with α = 1 (Eq. 11) | |
| P7 | | (Eq. S10) | | $r_{hyd}$ with α = 2 (Eq. 11) | |
| P8 | | | | | empirical[c] |

[a] Dry deposition reference height: $z_{GBC} \sim 60$ m, $z_{TNF} = 20$ m; $z_{HFEMS} = 29$ m.
[b] Mean Free Path (mfp) held constant across depositing trace gases.
[c] Empirical fit of non-stomatal cuticular deposition (Turnipseed et. al., 2006) modified herein for LAI.

## 355 3.1 Updates to the calculation of aerodynamic resistance

Table 3 contains an evaluation of parameterizations P1–P5 at the CTR site. Parameterization P1 overestimates daytime mean deposition velocities computed for the rapidly depositing species $H_2O_2$ (+15 %), HMHP (+41 %), and $HNO_3$ (+52 %). Nguyen et al. (2015) found excellent agreement between hourly GEOS-FP assimilated meteorology at this site (used herein for computation of $V_d$ in Table 3) and measured values, including $u_*$ and sensible and latent fluxes. GEOS-FP data report a

summertime $z_o = 2.2$ m for the 0.25° x 0.3125° grid cell that includes the CTR site—greater than would be expected at this site given the local 10 m canopy height. Prescribing $z_o$ to be 10 % of $h_c$ in parameterization P1b in accordance with conventionally used values for natural vegetation and in agreement with an updated land-use module developed for GEOS-Chem (Geddes et



al., 2016) results in a 35 % increase in $R_a$ and notable reductions in $V_d$ high biases. However, following the computation of $R_a$ in GEOS-Chem, P1b computes $R_a$ from a reference height of 60 m despite a measurement height of 22 m at the CTR site,

while neglecting to include a displacement height $d$. Neglecting $d$ from the computation of $R_a$ in Eq. (3) increases daytime $R_a$ in parameterization P1b by 1 % when referenced from 60 m and 9 % when referenced from 22 m (data not shown). Although the greatest sensitivity of $R_a$ to $z$ occurs in proximity to $z_o$, (Fig. S2), the difference between $R_a$ computed from an above-canopy measurement height vs. typical heights from which global CTMs reference dry deposition can be significant (Figs. S1 & S2). Referencing $R_a$ from the CTR measurement height of 22 m in parameterization P2 results in a 42 % decrease in $R_a$

under neutral conditions (Fig. S2) and a 23 % decrease under daytime (10–15 LST) conditions (Table 3, P1b vs P2). It should be noted that due to significant contributions of $R_b$ to the total resistance pathway for rapidly depositing species (Table 3, Section 3.2), referencing $R_a$ from GEOS-Chem grid-box-center instead of measurement height, as is commonly done in studies comparing deposition velocities from CTMs to measured values (Clifton et al., 2017; Nguyen et al., 2015; Nowlan et al., 2014; Silva and Heald, 2018), results in a moderate (8 %) decrease in $V_d$ for the species of Table 3 (P5, data not shown).

**Table 3:** Effects of updates to the calculation of aerodynamic resistance $R_a$ and quasi-laminar sublayer resistance $R_b$ on simulated daytime (10–15 LST) dry deposition velocities over Talladega National Forest (temperate, deciduous) for three rapidly depositing species. Serial modifications to base parameterization P1 are highlighted, i.e., PX (update). Shown are mean quantities ± standard deviations about the hourly timeseries[a].

| Parameterization[b] | | H₂O₂ | | | HMHP | | | HNO₃ | | |
|---|---|---|---|---|---|---|---|---|---|---|
| | $R_a$ | $R_b$ | $V_d$ | NMB[c] | $R_b$ | $V_d$ | NMB[d] | $R_b$ | $V_d$ | NMB[e] |
| | [s/m] | [s/m] | [cm/s] | [%] | [s/m] | [cm/s] | [%] | [s/m] | [cm/s] | [%] |
| P1 (base sim.) | 9.5±2.7 | 7.1±2.1 | 6.0±1.2 | 15 | 7.7±2.9 | 5.8±1.1 | 41 | 7.6±2.9 | 5.8±1.1 | 52 |
| P1b ($z_o = 0.1 h_c$)[f] | 12.8±3.4 | \| | 5.0±1.0 | -3 | \| | 4.9±1.0 | 19 | \| | 4.9±1.0 | 29 |
| P2 ($z_{ref}$ = 22 m - d) | 9.7±2.5 | \| | 5.9±1.3 | 14 | \| | 5.8±1.2 | 40 | \| | 5.8±1.2 | 51 |
| P3 (RSL, $u(z_o)$>0) | 10.2±2.6 | \| | 5.8±1.2 | 11 | \| | 5.6±1.2 | 36 | \| | 5.6±1.2 | 47 |
| P4 (RSL, $u(z_o)$=0) | 5.3±1.4 | \| | 7.9±1.7 | 52 | \| | 7.6±1.7 | 85 | \| | 7.6±1.7 | 100 |
| P5 (D update)[g] | \| | 12.9±4.9 | 4.4±1.0 | -15 | 14.9±5.7 | 4.1±1.0 | -1 | 15.7±6.0 | 4.0±0.9 | 4 |

[a] Mean quantities are averaged across the five daytime periods in June 2013 that Nguyen et al. (2015) use in their analysis of eddy covariance

observed deposition velocities (Section 2.2.1).

[b] Table 2 contains a list of parameterization updates. $R_c$ set to 1 s m⁻¹ following minimum allowed in GEOS-Chem (Section 2.1).

[c] To measured (eddy covariance) daytime (10–15 LST) $V_d(H_2O_2)$ = 5.2 ± 1.1 cm s⁻¹ (Nguyen, 2015).

[d] To measured (eddy covariance) daytime (10–15 LST) $V_d(HMHP)$ = 4.1 ± 1.1 cm s⁻¹ (Nguyen, 2015).

[e] To measured (eddy covariance) daytime (10–15 LST) $V_d(HNO_3)$ = 3.8 ± 1.3 cm s¹ (Nguyen, 2015).

[f] $z_o$ set to 10% of canopy height ($h_c$) for parameterizations P1b–P8.

[g] Parameterization P5 computes $R_a$ following P3.

Considering that the CTR and HFEMS measurement heights, ~ 2 $h_c$ and 1.5 $h_c$, respectively, are at the upper limits of the roughness sublayer (RSL), a region where turbulent mixing in the wake of roughness elements is enhanced above that

predicted by M–O similarity theory by a factor of 2 to 3 (Section S1.4), $R_a$ computed according to M–O similarity theory following Eq. (3) may be in slight underestimate due to non-zero horizontal winds at $z_o$ resulting from enhanced downward





mixing of momentum. To quantify this effect, parameterization P3 computes $R_a$ corrected for RSL mixing which allows $u(z_o)$ > 0 m s$^{-1}$ (Eq. (S10)), resulting in a small (5 %) increase in $R_a$ at the CTR measurement height under the daytime conditions of Table 3, and even smaller changes to $V_d$ given the influence of $R_b$ (Section 3.2). Given the lower relative measurement

height at the HFEMS (~ 1.5 $h_c$), P3 $R_a$ results in increases over P2 $R_a$ by 30 % (10[th] percentile), 20 % (50[th] percentile) and 18 % (90[th] percentile) at this site (Fig. S1); simulated $V_d(HNO_3)$ over Harvard Forest is discussed in Section 3.4.1. Parameterization P4 in Table 3 shows the effect of incorrectly neglecting the non-zero wind at $z_o$ in the RSL correction of $R_a$, resulting in a 50 % reduction in $R_a$ and a significant increase to the high biases in $V_d$ for the rapidly depositing species of Table 3. As demonstrated herein and in agreement with previous work (Simpson et al., 1998), it may be appropriate to neglect the

effects of the RSL on depositing species when referenced from a height of at least 1.5–2 $h_c$, however, studies endeavouring to understand bidirectional exchange or the dispersion of near-surface emissions should consider the effect of asymmetrical $R_a$ that the RSL imposes (Section S1.4).

**3.2 Updates to the calculation of molecular diffusivities**

As seen in Table 3, updates to the calculation of $R_a$ failed to address high biases in simulated deposition velocities of rapidly depositing species. Larger molecular weight species HMHP and HNO$_3$ exhibit a greater high bias in $V_d$, 36 % and 47 %, respectively, than the lower molecular weight species H$_2$O$_2$ (11 %). Given the dependence of maximum deposition velocity ($V_{d,max} = [R_a + R_b]^{-1}$) on molecular diffusivity $D$ through influence on $R_b$ (Eq. (4)) (Meyers et al., 1989), we evaluate the calculation of molecular diffusion coefficients in GEOS-Chem against measured values for atmospherically relevant

molecules.

   Figure 1 depicts a large high bias in calculated diffusion coefficients from the dry deposition module of GEOS-Chem, which employs the Chapman–Enskog theory for binary diffusivity (Seinfeld, 1986). The bias results from the use of a constant collision diameter $\sigma = 2.7$ Å for all species with air—an underestimate for many atmospherically relevant molecules, i.e., $\sigma$ for O$_3$ with air is 3.793 Å (Massman, 1998; Poling and Prausnitz, 2004). The collision diameter $\sigma$ is a pairwise characteristic

length scale of the Lennard–Jones intermolecular force, which is not readily available for many atmospheric trace gasses (Tang et al., 2014). Several semi-empirical methods have been proposed for the estimation of $D$ in low pressure binary systems (Poling and Prausnitz, 2004). Fuller et al. (1966) developed a simple and generalized semi-empirical correlation equation for the estimation of binary gas phase diffusion coefficients using additive atomic diffusion volumes $V_i$ for each species $\sum_A V_i$ and $\sum_B V_i$. The diffusion coefficient $D$ [cm$^2$ s$^{-1}$] for trace gas $A$ in bath gas $B$ is given by:

$$D = \frac{10^{-3} T^{1.75} \left( 1/M_A + 1/M_B \right)^{1/2}}{P \left[ (\Sigma_A V_i)^{1/3} + (\Sigma_B V_i)^{1/3} \right]^2},\qquad(8)$$

where $P$ is the pressure [atm], $T$ is the temperature [K], and $M$ is the molecular mass [g mol$^{-1}$]. Atomic, and in some cases molecular, diffusion volumes were obtained from regression analysis of 153 binary systems across 340 T-P states and are





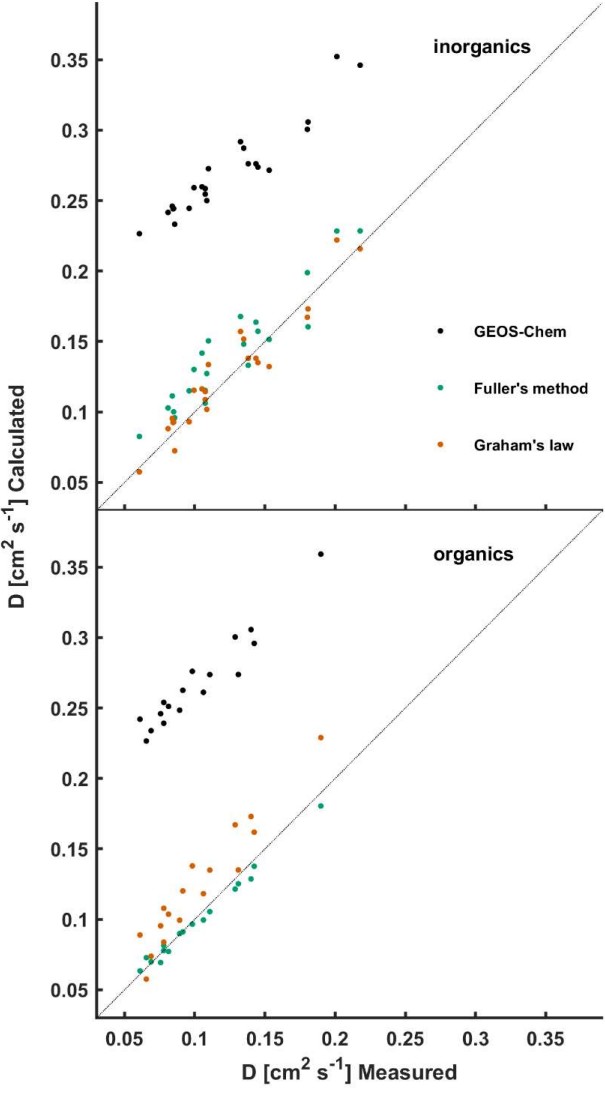

**Figure 1:** Measurements of gaseous diffusion coefficients of atmospherically relevant molecules in air or N$_2$ at STP are compared to calculated values. Diffusivities calculated following the method used in GEOS-Chem are compared to those calculated following Fuller's method and Graham's law (referenced from $D_{CO_2}$). Measured and computed (Fuller's method) values are listed in Table S1 in the supplement.



summarized in Poling et al. (2004), Tang et al. (2014), and Tang et al. (2015). As seen in Fig. 1, diffusion coefficients computed

using Fuller's method result in a much-improved comparison to measurements, with better agreement to organic species ($R^2$

= 0.99 and NMB = -3 %) than to inorganics ($R^2$ = 0.88 and NMB = 13 %), consistent with the findings of Tang et al. (2014 &

2015) from an evaluation of a comprehensive diffusivity dataset of atmospherically relevant reactive trace gases for which

Tang et al. have made the results publicly available.

435         Figure 1 also depicts molecular diffusion coefficients approximated by Graham's law of effusion, i.e., $D_{1k} =$

$D_{2k}\sqrt{M_2/M_1}$ (Mason and Evans, 1969), where (continuum) diffusion coefficients are approximated by Knudsen diffusion

coefficients $D_k$— an oversimplification of Eq. (8), albeit, a strategy used in the atmospheric science community nonetheless

(Nguyen et al., 2015; Weber and Renenberg, 1996; Wesely, 1989). Resulting diffusion coefficients scaled from measured $D_{CO_2}$

correlate well with measured values ($R^2$ = 0.91), with NMB to inorganic and organic species of 3 % and 20 %, respectively.

In a review of molecular diffusivities of atmospherically relevant molecules, Massman et al. (1998) note misapplication of

Graham's law to molecular diffusivities can lead to errors of up to 23 %. Referencing Graham's law from measured $D_{H_2O_2}$, as

done in Nguyen et al. (2015), degrades comparison to inorganic diffusivities (NMB ~ -14 %), improves comparison to organics

(NMB < 2 %), and has no effect on correlation ($R^2$ = 0.91) (data not shown). Sensitivity of Graham's law to choice of reference

species is not surprising given the deviation of the $\sqrt{M_2/M_1}$ dependence from the functional form of Eq. (8). Measured and

computed (Fuller's method) diffusion coefficients presented in Fig. 1 are tabulated in Table S1. We do not differentiate

between diffusivity measurements carried out in air or $N_2$, as differences are expected to be small, i.e., a 2 % difference in $D_{O_3}$

at STP in air vs $N_2$ according to Fuller's method. We assume an air bath gas for all diffusion coefficients computed via Fuller's

method.

        Parameterization P5 computes $R_b$ according to Eq. (4) using measured diffusion coefficients when available and

diffusion coefficients according to Fuller's method in the absence of measured values. Diffusion coefficients are adjusted to

ambient T-P following Eq. (6) prior to use in Eq. (4). Eliminating the high bias in calculated molecular diffusivities resulted

in a near doubling of $R_b$ for the species in Table 3, and a much-improved comparison to the daytime deposition velocities of

the larger molecular weight species HMHP (NMB -1 %) and $HNO_3$ (NMB 4 %). The increase in $R_b$ for $H_2O_2$ results in a low

bias of -15 %, but well within the large relative uncertainty for $R_b$ due to variations in canopy structures (Massman, 1994;

Sievering et al., 2001).

        Molecular diffusivity is also involved in the calculation of $R_c$ via influence on stomatal resistance $r_s$, which is scaled

by the ratio $D_{H_2O}/D_x$ in dry deposition parameterizations commonly used in chemical transport models (Wesely, 1989; Zhang

et al., 2003a). The effect of updated molecular diffusivity on $R_c$ in GEOS-Chem is significant for molecules which dry deposit

under stomatal control, i.e., species with low aqueous solubility or surface reactivity, and is discussed in Section 3.4.1.

460         As noted by Nguyen et al. (2015), the practice of setting $V_d$ for rapidly depositing species equal to $V_d(HNO_3)$ neglects

species-specific diffusion limitations, which can be important under turbulent conditions when $R_a$ is at a minimum. For



example, $R_b$ for isoprene nitrate is estimated to be 23 % greater than for $HNO_3$, translating to a -12 % bias in $V_d$ under the median midday conditions at the HFEMS ($R_a(60 m)$ = 8.6 s m$^{-1}$, $R_b(HNO_3)$ = 12 s m$^{-1}$; Fig. S6).

**3.3 Nocturnal dry deposition of $NO_2$ over Harvard Forest**

**3.3.1 Eddy covariance observed $V_d(NO_2)$**

Nocturnal hourly eddy covariance $NO_2$ fluxes and resulting exchange velocities $V_{ex}(NO_2)$ over Harvard Forest from April–November 2000 are shown in Fig. 2 as a function of $NO_2$ concentration. We restrict our analysis to nighttime (20–04 LST), when above-canopy $NO_2 : NO_x \sim 1$ and photochemical flux divergence of the $NO-NO_2-O_3$ triad due to the presence of a vertical gradient in irradiance through the forest canopy (Gao et al., 1993) is absent. As seen in the top panel of Fig. 2, nocturnal fluxes

of $NO_2$ over Harvard Forest are predominantly (~ 70 %) downward, especially at higher ambient $NO_2$ concentrations. Nocturnal mean (median) ± 1σ fluxes of $NO_2$ from April through November are -0.8 (-0.3) ± 2 ppb cm s$^{-1}$. These downward (p < 0.01) above-canopy aggregate fluxes of $NO_2$ are comparable in magnitude to counteracting summertime nocturnal soil NO emissions, estimated by Munger et al. (1996) through a mass-balance approach to be ~ 0.9 μmol m$^{-2}$ h$^{-1}$ (3.5 ng N m$^{-2}$ s$^{-1}$, or 0.62 ppb cm s$^{-1}$) at the HFEMS. Munger et al. (1996) note that nocturnal NO is elevated near the forest floor and Horii et

al. (2004) find decreasing within-canopy nocturnal NO profiles at Harvard Forest with above-canopy concentrations and fluxes indistinguishable from zero despite net downward fluxes of $NO_2$, presumably due to titration of soil-emitted NO by $O_3$ on a timescale much shorter (minutes) than in-canopy vertical mixing, followed by nocturnal canopy loss processes for $NO_2$. Studies have noted the importance of knowledge of local soil NO emissions and within-canopy processes involving $NO_x$ when interpreting above-canopy $NO_2$ fluxes (Delaria and Cohen, 2020; Eugster and Hesterberg, 1996; Flechard et al., 2011; Min et

al., 2014). Using measured soil NO emissions from a Ponderosa Pine plantation 75 km from Sacramento, California, Min et al. (2014) calculate an $NO_2$ flux resulting from the reaction of soil NO with $O_3$ to be 3.5 times greater than the observed above-canopy eddy covariance $NO_2$ flux, indicating in-canopy $NO_2$ loss processes which authors mostly attribute to daytime organic nitrate production. In their analysis of eddy covariance fluxes of $NO_2$ over a managed grassland in central Switzerland, Eugster & Hesterberg (1996) found that accounting for counteracting fluxes of soil-emitted NO, oxidized to $NO_2$ below the height of

the sensor (~ 2.7 to 3.6 ng N m$^{-2}$ s$^{-1}$), resulted in an increase in inferred nocturnal $V_d(NO_2)$ by up to a factor of 2; sensitivity tests showed a 50 % change in estimated soil NO emission resulted in a change in inferred $V_d(NO_2)$ on the order of 25 %.

In an effort to isolate the contribution that dry deposition makes to above-canopy nocturnal eddy covariance fluxes ($F_{EC}$) of $NO_2$, we infer $V_d(NO_2)$ following Eq. (9) to account for the effects of nocturnal chemical flux divergence ($V_{chem}$) and counteracting soil NO emissions assumed to rapidly titrate with $O_3$ and ventilate the canopy as $NO_2$ ($F_{soil}$). The resulting

$V_d(NO_2)$ is a best estimate of the nocturnal dry deposition pathway with which to evaluate parameterizations:

$$V_d + V_{chem} = -V_{ex} = -\frac{(F_{EC} - F_{soil})}{[NO_2]}, \tag{9}$$



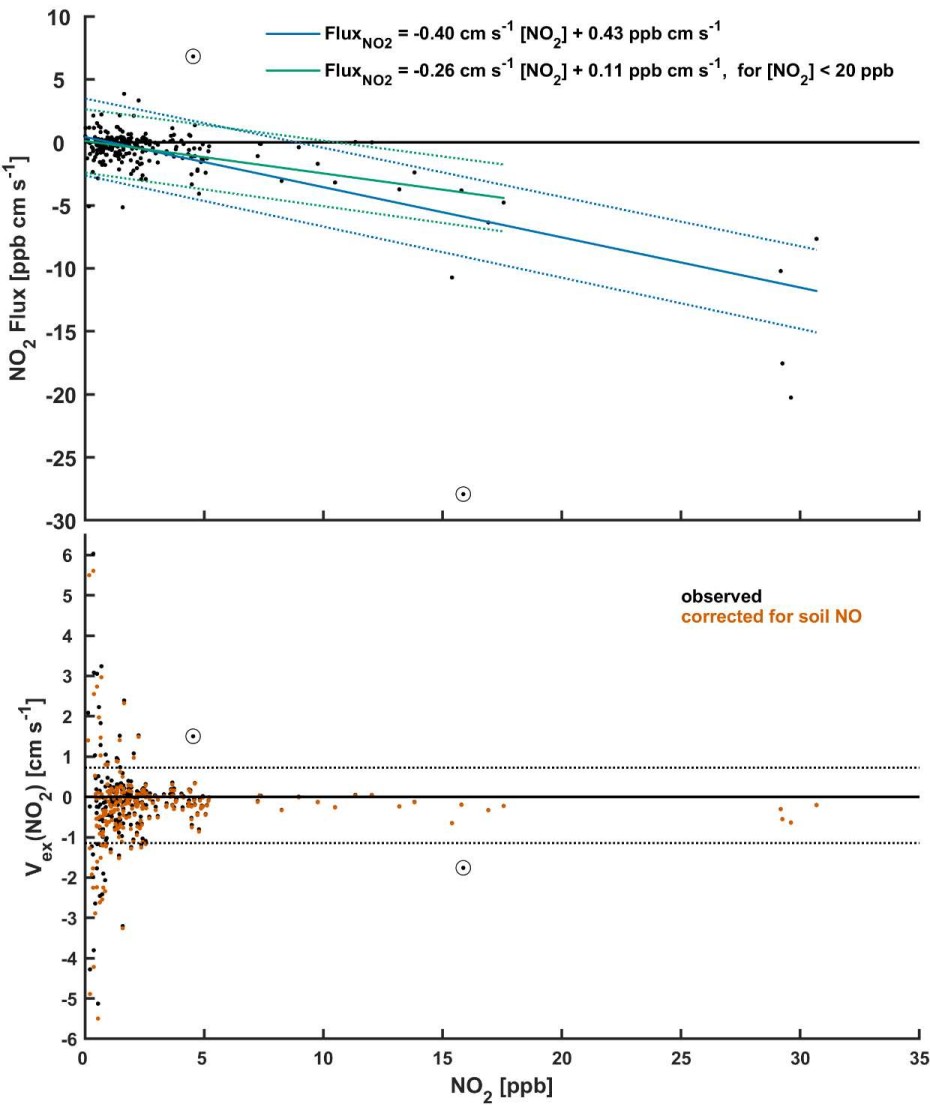

**Figure 2:** Nocturnal (20–04 local solar time) hourly eddy covariance NO$_2$ fluxes (**TOP**) and resulting exchange velocities $V_{ex}(NO_2)$ (**BOTTOM**) as a function of NO$_2$ concentration. These publicly available measurements (Horii, 2004) were taken over an established mixed deciduous forest (Harvard Forest, MA, U.S.) from April–November 2000. Estimated above-canopy soil NO flux was subtracted from measured hourly NO$_2$ fluxes in order to estimate $V_d(NO_2)$ due to deposition (depicted as 'corrected for soil NO'; Eq.(9)). Included in the top plot are linear fits and associated 95% prediction intervals. Dashed lines in the bottom plot depict boundaries of an outlier filter applied to hourly $V_{ex}(NO_2)$ prior to calculation of means (Section 2.2.2). Data points excluded from analysis based on visual inspection are circled. Hourly observations made under conditions of low turbulence ($u_* < 0.2$ m s$^{-1}$) were excluded from analysis.



where $V_{ex}$ is the eddy covariance observed $NO_2$ exchange velocity which does not assume predominant deposition and therefore has sign convention analogous to $F_{EC}$; $V_{chem}$ represents an estimate of below-sensor nocturnal chemical loss of $NO_2$ via formation and loss of $N_2O_5$, limited by the rate of oxidation of $NO_2$ with $O_3$ (Browne and Cohen, 2012; Jacob, 2000). We use an estimate of the maximum rate of nocturnal chemical loss of $NO_2$ proposed by Horii et al. (2002) in their analysis of the dataset used herein, $V_{chem} \sim 0.05$ cm s$^{-1}$, which translates to a below-sensor (< 29 m) nocturnal chemical lifetime of $NO_2$ to

oxidation by $O_3$ of ~ 16 h. The bottom panel of Fig. 2 includes hourly values of $V_{ex}(NO_2)$ corrected for soil NO. Values of $F_{soil}$ used in Eq. (9) are less than the summertime forest floor estimate from Munger et al. (1996) due to seasonality and within-canopy loss processes. Hourly estimates of $F_{soil}$ were calculated by scaling the reported summertime nocturnal soil NO emission flux at Harvard Forest $F_{NO,summer}$ (0.62 ppb cm s$^{-1}$) by GEOS-Chem simulated seasonality $\kappa$ and a parameterized canopy reduction factor $CRF$:

$F_{soil}(hr) = F_{NO,summer}\ \kappa(month)\ [1 - CRF(hr)] ,$                                         (10)

Month-specific $\kappa$ scale factors were obtained by normalizing simulated monthly mean nocturnal soil NO emission, output at the location of the HFEMS from a high resolution ($0.25°$ x $0.3125°$) GEOS-Chem simulation, by the peak monthly mean simulated emission (July at the location of HFEMS). GEOS-Chem simulated soil NO emission in the region of Harvard Forest exhibits significant seasonality, with winter minimum a small fraction (< 5 %) of the summertime maximum (Fig. S3). Section

S3 in the supplement describes the parameterization of CRF used in GEOS-Chem and herein in Eq. (10).

       Figure 3 depicts monthly nocturnal $V_d(NO_2)$ observed over the HFEMS from April–November 2000 alongside coincidently-sampled simulated values from parameterizations P5–P7. Table 4 presents observed and simulated values of $V_d(NO_2)$ aggregated across all months, as well as associated canopy reduction factors used to correct observed $V_d(NO_2)$ for soil NO. We begin discussion of eddy covariance observed bulk-canopy $V_d(NO_2)$ below, followed by discussion in Section 3.3.2

of the bias in simulated values stemming from the widely used W89 parameterization of surface resistances. Mechanistic updates to the parameterization of $NO_2$ surface uptake (P6–P7) developed to remedy large biases in nocturnal $V_d(NO_2)$ computed following the W89 scheme (P5) are discussed in Section 3.3.3.

       As previously mentioned, hourly values of observed $V_{ex}(NO_2)$ were subjected to an outlier filter (Fig. 2) prior to computation of mean values, whereas median and 'mean flux–to–mean concentration' ratios

( $\bar{F}/\overline{[NO_2]}$ ) included in Table 4 were not, and instead computed directly from u$^*$ filtered (> 0.2 m s$^{-1}$) hourly data as the latter two statistics are less influenced by outliers than arithmetic means. Aggregate values of $\bar{F}/\overline{[NO_2]}$ in Table 4 are in the same units as $V_d(NO_2)$ [cm s$^{-1}$] and include corrections for $F_{soil}$ and $V_{chem}$ as do mean and median quantities computed from hourly values following Eq. (9). Assuming first-order dependence of $NO_2$ dry deposition with concentration (Eq. (1)), computing values of $\bar{F}/\overline{[NO_2]}$ over long averaging times is a strategy to reduce the influence of random variability in deposition velocity

estimates, especially under low $NO_2$ conditions as evident in Fig. 2. Although we report biases between simulated and observed $V_d(NO_2)$ using outlier-filtered mean values, median and $\bar{F}/\overline{[NO_2]}$ values of $V_d(NO_2)$ are included in Table 4 for comparison.




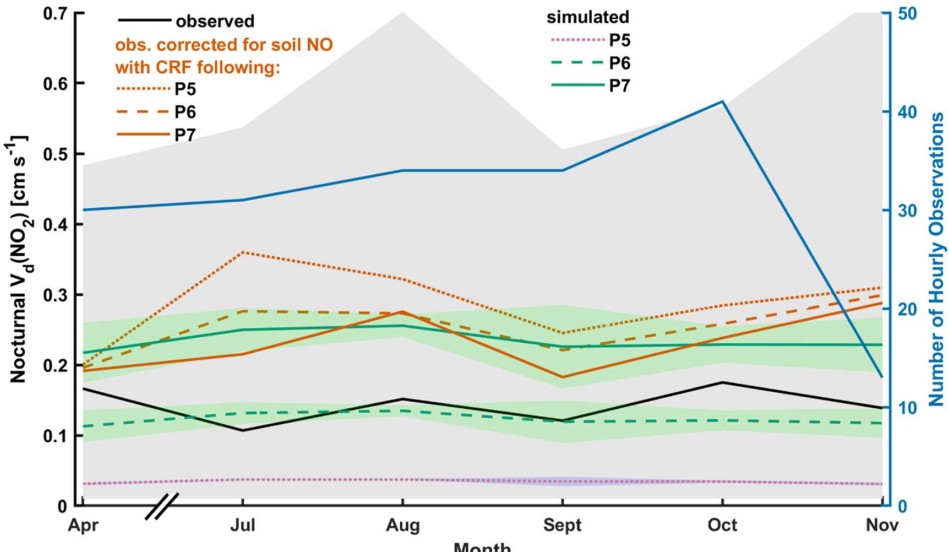

**Figure 3:** Observed and simulated monthly mean nocturnal (20–04 local solar time) $NO_2$ deposition velocities $V_d(NO_2)$ over Harvard Forest. Simulated values are coincidently sampled with hourly observations prior to averaging. Also depicted is observed $V_d(NO_2)$ corrected for soil
NO emission using simulated soil NO canopy reduction factors (CRF) from parameterizations P5, P6, and P7. Standard deviation about simulated monthly mean values, as well as measured monthly mean values corrected for soil NO using CRF from P7, are depicted as shaded areas. Insufficient data prevented analysis for May and June (Fig. S7).

As seen in Fig. 3, monthly mean values of observed $V_d(NO_2)$ uncorrected for the influence of soil-emitted NO are in the range of 0.1–0.2 cm s$^{-1}$. Although variability in observed nocturnal $V_d(NO_2)$ is large, with standard deviations greater than

mean values, corrections for soil NO (venting the canopy as $NO_2$) results in a significant ($p < 0.03$) increase in nocturnal $V_d(NO_2)$, yielding monthly mean values in the approximate range of 0.2–0.3 cm s$^{-1}$. Both uncorrected and soil NO corrected nocturnal $V_d(NO_2)$ lack discernible seasonality. In Section 3.3.4, bottom-up estimates of nocturnal $V_d(NO_2)$ for Harvard Forest are developed in an effort to understand the apparent lack of seasonality in top-down observations. Large variability in eddy covariance observed $NO_2$ flux and resulting deposition velocities have been noted in other studies (Eugster and Hesterberg,

1996; Farmer et al., 2006; Geddes and Murphy, 2014), wherein authors restrict analysis to average values in order to reduce the variability in these complex ecosystem-scale observations (Baldocchi, 2003). Herein, we restrict analysis to average values over at least one month, as simple resistance-in-series parameterizations of dry deposition employing 'big leaf' representations of $R_c$ are designed for computational expediency, general applicability over a wide range of land types, and to reflect average estimates over weeks to months and therefore lack the necessary complexity to capture the full range of short-term variability

at specific sites (Wesely, 1989). Our objective is to address potential long-term biases in the parameterization of non-stomatal $V_d(NO_2)$ in GEOS-Chem, noted to be significant by Horii et al. (2002) in their initial presentation of this dataset.



**Table 4:** Nocturnal (20–04 local solar time) NO₂ deposition velocities over Harvard Forest aggregated from April–November. Measured values, with and without estimated soil NO corrections using simulated canopy reduction factors (CRF) corresponding to parameterizations P5–P7, are shown along with coincidently sampled simulated values. Measurements under conditions of low turbulence (friction velocity $u_*$ < 0.2 m s⁻¹) were excluded from analysis, leaving 230 hourly observations in the timeseries (Section 2.2.2).

| | CRF | $V_d(NO_2)$ [cm s⁻¹] | | |
| --- | --- | --- | --- | --- |
| | [%] | mean[a] | median | $\bar{F}/\overline{[NO_2]}$ [b] |
| **Measured** | | | | |
| no soil NO | N/A | 0.15 ± 0.34 (185) | 0.13 | 0.21 |
| soil NO, CRF(P5) | 31 | 0.28 ± 0.35 (181) | 0.25 | 0.30 |
| soil NO, CRF(P6) | 47 | 0.25 ± 0.34 (181) | 0.22 | 0.28 |
| soil NO, CRF(P7) | 59 | 0.23 ± 0.35 (183) | 0.21 | 0.26 |
| **Simulated[c]** | | | | |
| P5 ($R_a$ & $D$) | N/A | 0.04 ± <0.01 | 0.04 | - |
| P6 ($\alpha = 1$) | N/A | 0.12 ± 0.02 | 0.13 | - |
| P7 ($\alpha = 2$) | N/A | 0.24 ± 0.04 | 0.24 | - |

[a] Measured hourly values of $V_{ex}(NO_2)$ were subjected to an outlier filter (Fig. 2) prior to computing mean $V_d(NO_2)$; remaining number of hourly observations are included in brackets adjacent corresponding mean values and standard deviations.
[b] The ratio of 'mean NO₂ flux-to-mean NO₂ concentration' ($\bar{F}/\overline{[NO_2]}$) in units of [cm s⁻¹] is included for comparison (Section 3.3.1).
[c] See Table 2 for serial updates. Briefly,
  P5: updates to aerodynamic resistance ($R_a$) and molecular diffusivity ($D$).
  P6 : update resistance to surface uptake of NO₂ ($R_c(NO_2)$) to include reaction R1 using surface area scale factor $\alpha = 1$ (Eq.(11)).
  P7 : Analogous to P6 but with surface area scale factor $\alpha = 2$.

### 3.3.2 Evaluation of parameterized $V_d(NO_2)$ from GEOS-Chem

Parameterization P5, which computes NO₂ surface uptake resistance $R_c(NO_2)$ following the W89 representation in GEOS-Chem, yields a simulated nocturnal $V_d(NO_2)$ that is biased low by nearly 4-fold compared to observations uncorrected for soil NO, increasing to a 7-fold low bias after correcting for soil NO with a corresponding simulated CRF from parameterization P5 of 31% (Table 4). This underestimate is driven by the large nocturnal $R_c(NO_2)$ of ~ 2,700 s m⁻¹ in parameterization P5 (Fig. S6), which has been noted in previous studies comparing NO₂ dry deposition simulated by the W89 algorithm to eddy covariance observations over forest (Horii, 2002) and grassland (Eugster and Hesterberg, 1996) ecosystems. In particular, Eugster & Hesterberg (1996) inferred a median value for nocturnal non-stomatal $R_c(NO_2)$ of 700 s m⁻¹ (range 500–950 s m⁻¹) over a managed grassland in central Switzerland—a surface resistance on the order of 4 times lower than predicted by the W89 algorithm.

### 3.3.3 Updates to parameterized $V_d(NO_2)$ by representing NO₂ hydrolysis on deposition surfaces

Horii et al. (2002) note that observed nocturnal dry deposition of NO₂ may result from a surface hydrolysis reaction following R1. In parameterizations P6–P8 we replace the non-stomatal components of the bulk-surface resistance scheme for NO₂ with



a dry deposition pathway representing $NO_2$ hydrolysis $r_{hyd}$ [s m$^{-1}$], formulated as a collision-limited heterogeneous reaction with ground surfaces (Cano-Ruiz et al., 1993):

$$r_{hyd} = \frac{4}{\gamma_{g,NO_2} \, \overline{v_t} \, \alpha},$$  (11)

where $\gamma_{g,NO_2}$ is a ground uptake coefficient for $NO_2$ resulting from heterogeneous hydrolysis on deposition surfaces, $\overline{v_t}$ the mean thermal speed of $NO_2$, and $\alpha$ a dimensionless scale factor introduced herein to facilitate application of Eq. (11) across land types of varying surface area densities. Lammel et al. (1996) recommend reaction R1 be parameterized in atmospheric chemistry models using field-derived uptake coefficients, as realistic conditions, in particular surface area densities, are difficult to reproduce in the lab. We employ the field-derived ground uptake coefficient $\gamma_{g,NO_2}$ for reaction R1 from

VandenBoer et al. (2013), determined from resulting HONO emitted into the nocturnal boundary layer over a wintertime agricultural region in Colorado, USA. Consistent with the heterogenous hydrolysis of $NO_2$ requiring adsorbed water to proceed, VandenBoer et al. (2013) parameterize $\gamma_{g,NO_2}$ as a function of RH [%] according to Eq. (12) to capture the factor of 2 variability in $\gamma_{g,NO_2}$ on either side of their best fit value (8 x 10$^{-6}$):

$$\gamma_{g,NO_2} = \frac{RH}{50} \, 8 \, x \, 10^{-6},$$  (12)

590   Parameterization P6 computes $r_{hyd}$ with $\alpha = 1$, resulting in a simulated nocturnal mean $V_d(NO_2)$ of 0.12 ± 0.02 cm s$^{-1}$—a 3-fold increase over P5 and satisfactory agreement with observed mean $V_d(NO_2)$ uncorrected for soil-emitted NO, however, underestimating soil NO corrected observations by ~ 50 % (Fig. 3 & Table 4). The larger nocturnal mean CRF of 47 % for parameterization P6 is due to reduced nocturnal $R_c(NO_2)$ (median value ~ 750 s m$^{-1}$, Fig. S6), resulting in a small (11 %) decrease in observed mean $V_d(NO_2)$ corrected for soil NO. Increasing the rate of non-stomatal uptake of $NO_2$ by computing

$r_{hyd}$ with $\alpha = 2$ in parameterization P7 resulted in an unbiased simulation of nocturnal mean $V_d(NO_2)$ of 0.24 ± 0.04 cm s$^{-1}$ compared to observed mean $V_d(NO_2)$ of 0.23 ± 0.35 cm s$^{-1}$ after correction for soil-emitted NO using the CRF from P7 (nocturnal mean CRF of 59 %, Table 4). Although parameterization P7 results in satisfactory simulation of nocturnal $V_d(NO_2)$ at the HFEMS when averaged across all months (Table 4), intra- and inter-month variability in observed $V_d(NO_2)$ is not captured in simulated values (Fig. 3).

600   Physical justification for the scale factor value $\alpha = 2$ necessary to match simulated with observed nocturnal mean $V_d(NO_2)$ could stem from a larger surface area available for $NO_2$ heterogeneous hydrolysis in a mature forest environment compared to the U.S. Midwest wintertime agricultural region over which VandenBoer et al. (2013) derived $\gamma_{g,NO_2}$. Heterogeneous reactions not limited by transport or diffusion to reaction surfaces are governed by a collision-limited rate which scales linearly with the surface area–to–volume ratio of the reaction vessel or environment (Jacob, 2000). Heterogeneous

hydrolysis of $NO_2$ may proceed on any surface accommodating adsorbed water, including foliar surfaces, bark, or elements of the forest floor (i.e., rock, soil, and debris). Despite the hydrophobic nature of many foliar surfaces, thin aqueous films have been observed on coniferous needles (Altimir et al., 2006; Burkhardt and Gerchau, 1994) and stomata-bearing surfaces of



deciduous leaves (Burkhardt et al., 1999) at ambient humidities well below saturation. In addition to radiative cooling, elevated humidity within the thin laminar boundary layer surrounding leaves may result from stomatal transpiration (Burkhardt and Hunsche, 2013) and to a lesser extent the hydraulic activation of stomata (HAS) where liquid water is drawn from sub-stomatal cavities along stomatal walls to deposited hygroscopic material on the leaf surface (Burkhardt, 2010), a process discussed further in Section 4 and S4. Surface area indices [$m^2\ m^{-2}$] for forest components at the HFEMS have been estimated (Fig. S4), including for stems and twigs (STAI = 0.9), coniferous needles (CAI = 0.8), and deciduous leaves (DLAI = 3.4 summertime maximum). Assuming round stems and twigs (Sörgel et al., 2011) and oblate coniferous needles (Oren et al., 1986; Riederer et al., 1988), the total wintertime canopy surface area is estimated as $\pi STAI + 2.7 CAI \sim 5\ m^2\ m^{-2}$. We estimate the summertime canopy surface area to be $\sim 12\ m^2\ m^{-2}$ accounting for both sides of deciduous leaves or $\sim 9\ m^2\ m^{-2}$ neglecting the non-stomatous adaxial (top) surface of deciduous leaves, in agreement with typical macroscopic surface area indices for temperate and boreal forest canopies of $12\ m^2\ m^{-2}$ (range 5–14 $m^2\ m^{-2}$) (Lammel, 1999). The surface area of the forest floor, including debris, would also be much greater than the planar ground area, and that of tree bark greater than the simple geometric surface area (discussed further in Section 3.3.4).

The lack of seasonality in observed nocturnal $V_d(NO_2)$ depicted in Fig. 3 may reflect an interseasonal buffering of available surface area for reaction of above-canopy $NO_2$ due to increased air parcel mixing throughout the lower canopy in the absence of deciduous leaves (see Section 3.3.4). We did not attempt to parameterize non-stomatal deposition of $NO_2$ to upper and lower canopy elements separately in our top-down optimization of $R_c(NO_2)$, as is currently the approach in the W89 and Z03 dry deposition schemes. Due to the lack of discernible seasonal variability in observed nocturnal $V_d(NO_2)$, above-canopy observations were insufficient to justify the additional variables. We acknowledge that the nocturnal canopy environment to which we optimize simulated $V_d(NO_2)$ is under reduced turbulent mixing compared to daytime conditions when the forest would experience enhanced vertical exchange (Bannister et al., 2022; Sörgel et al., 2011; Thomas and Foken, 2007). Although daytime surface area available to above-canopy deposition is therefore likely to be greater than at night, nighttime sensitivity of $V_d(NO_2)$ to $\alpha$ is much greater than during the day when stomata are open and foliar uptake of $NO_2$ is a more substantial pathway to deposition than non-stomatal uptake (Fig. S6). Increasing $\alpha$ from 1 to 2 results in a 100 % increase in simulated $V_d(NO_2)$ at night (Table 4), but only a 10 % increase during mid-day (discussed further in Section 3.4.1).

The canopy compensation point for $NO_2$ is the ambient above-canopy concentration at which point consumption (i.e., dry deposition) and production (i.e., soil emission) are in balance (Duyzer et al., 1995). Studies of above-canopy $NO_2$ exchange have observed aggregate fluxes to be upward (Min et al., 2014; Vaughan et al., 2016), downward (Coe and Gallagher, 1992; Horii et al., 2004; Walton et al., 1997), and not significantly different from zero (Geddes and Murphy, 2014)—highlighting the importance of knowledge of below-canopy $NO_x$ emission and subsequent uptake and reaction in the interpretation of above-canopy fluxes. Although foliar compensation points for $NO_2$—a concentration below which vegetation was proposed to become a net source of $NO_2$—have been observed in the past via leaf-level chamber measurements to be generally < 2 ppb (Geßler et al., 2002; Sparks et al., 2001; Weber and Renenberg, 1996), recent chamber studies employing highly specific $NO_2$ detection methods have failed to observe foliar emission (Breuninger et al., 2013; Chaparro-Suarez et al., 2011; Delaria et al.,



2020, 2018; Wang et al., 2020). Although chamber studies generally observe first-order uptake of $NO_2$ under controlled conditions, constant $V_d(NO_2)$ inferred from linear regression of eddy covariance fluxes of $NO_2$ versus concentration is not expected due to variability in turbulence and surface conditions affecting uptake (i.e., surface wetness, stomatal aperture, and surface area). By restricting analysis to nocturnal conditions when RH is generally high (Fig. S5), stomata assumed closed (Section 4), and turbulence well established ($u^* > 0.2$ m s$^{-1}$), we find monthly aggregate $V_d(NO_2)$ relatively constant in the range 0.2–0.3 cm s$^{-1}$ across April–November, with expected large variability on finer timescales (i.e., $\sigma = 0.35$ cm s$^{-1}$ across the hourly dataset). Linear regression of hourly nocturnal $NO_2$ flux versus ambient $NO_2$ concentration (Fig. 2) yields a $V_{ex}(NO_2)$ of -0.40 cm s$^{-1}$ (p < 0.01) over the entire $NO_2$ concentration range (up to ~ 30 ppb) and -0.26 cm s$^{-1}$ (p < 0.01) when the four outlying hourly observations beyond 20 ppb $NO_2$ are excluded—consistent with the findings of Horii et al. (2004). An inferred $V_d(NO_2)$ of 0.21 cm s$^{-1}$ is obtained from the fit $V_{ex}(NO_2)$ value of -0.26 cm s$^{-1}$ from Fig. 2 after subtraction of $V_{chem} = 0.05$ cm s$^{-1}$—similar to aggregate values presented in Table 4. The y-axis intercept of 0.11 ppb cm s$^{-1}$ in Fig. 2, although not significant (p > 0.1), is in line with the estimated mean (April–November) above-canopy $NO_2$ flux of 0.13 ppb cm s$^{-1}$ resulting from soil NO emission and an average CRF of 59% from parameterization P7. An empirical CRF of ~ 70 % is obtained from the ratio of y-axis intercept (0.11 ppb cm s$^{-1}$, Fig. 2) to seasonal mean below-canopy soil NO flux (0.39 ppb cm s$^{-1}$). An $NO_2$ canopy compensation point for Harvard Forest, likely due to soil NO emission, is approximated by the x-axis intercept of ~ 0.4 ppb in Fig. 2.

By replacing the non-stomatal pathways of $NO_2$ deposition from the W89 algorithm with $r_{hyd}$ according to Eq. (11), we assume that non-stomatal deposition of $NO_2$ is due entirely to heterogeneous hydrolysis following reaction R1. Zhang et al. (2003) neglect solubility contributions to $NO_2$ uptake in their dry deposition scheme, relying entirely on similarity to $O_3$ reactivity. The W89 dry deposition scheme assigns $NO_2$ to the 'slightly reactive' category, intended for substances with limited biological reactivity but still requiring very small leaf mesophyll resistances so that $NO_2$ deposits under stomatal control. This classification for $NO_2$ in the W89 scheme results in near-negligible non-stomatal deposition, yielding a non-stomatal $R_c(NO_2)$ of ~ 2,700 s m$^{-1}$ at Harvard Forest—much above observation-inferred values, as previously discussed. Adding $r_{hyd}$ in parallel to the W89 non-stomatal deposition pathway, instead of in replacement of, results in a slight increase (~ 10 %) in simulated mean $V_d(NO_2)$ over Harvard Forest for parameterization P7—still supporting P7 with $\alpha > 1$, but possibly not as large as $\alpha = 2$. Variability in observed $V_d(NO_2)$ and uncertainties in the assumption of a non-stomatal $R_c(NO_2)$ pathway following similarity to $SO_2$ and $O_3$ uptake make more precise recommendations difficult.

Given the dependence of $r_{hyd}$ on surface area, land type specific $\alpha$ values evaluated across seasons would be desirable to improve confidence for use in global CTMs. As previously mentioned, Eugster & Hesterberg (1996) inferred a nocturnal non-stomatal median value for $R_c(NO_2)$ of 700 s m$^{-1}$ (range 500–950 s m$^{-1}$) over a managed grassland in central Switzerland from soil NO corrected eddy covariance observations—similar to the median value of 750 s m$^{-1}$ simulated herein following Eq. (11) with $\alpha = 1$ (Fig. S6). Pilegaard et al. (1998) report nocturnal $R_c(NO_2)$ of 771 ± 111 s m$^{-1}$ inferred from eddy covariance observations over a harvested wheat field (with re-growth) in southern Germany during mid-September. Although



soil NO contributions to above-canopy $NO_2$ flux were not considered in their analysis, given the high nocturnal $NO_2$ concentrations of 10–30 ppb at the location, soil NO most likely had a reduced relative effect on resulting $V_d(NO_2)$ compared to the large influence noted by Eugster & Hesterberg (1996) where nocturnal $NO_2$ concentrations were less than 10 ppb during periods when soil NO emission occurred (T > 5 °C). Coe et al. (1992) used eddy covariance to estimate a non-stomatal $R_c(NO_2)$ of 548 s m$^{-1}$ over a Heather moorland located in southern Netherlands. Plake et al. (2015) find a maximum median nocturnal

bulk $R_c(NO_2)$ over a natural grassland site in Mainz, Germany (August–September) of 560 s m$^{-1}$ via the dynamic chamber approach, attributing all flux of $NO_2$ to deposition since soil NO emissions for this nutrient poor site were below chamber detection limits. The nocturnal $R_c(NO_2)$ values reported by Coe et al. (1992) and Plake et al. (2015) are intermediate between $r_{hyd}$ values computed using α = 1 & 2. Assigning α = 1 for low roughness vegetative land types appears reasonable but may yield slight underestimates in nocturnal $NO_2$ uptake under some conditions.

685        Reaction R1 has been observed to proceed efficiently on ice surfaces, even at low temperatures (< 170 k) (Bang et al., 2015; Kim and Kang, 2010). Stocker et al. (1995) observed via the eddy covariance technique nocturnal deposition of $NO_2$ to a snow-covered grassland in northern Colorado, reporting a median resistance to surface uptake of 740 s m$^{-1}$—similar to $r_{hyd}$ of 725 s m$^{-1}$ following Eq. (11) with α = 1, mean thermal speed ($\bar{v}_t$) computed at 260 K, and $\gamma_{g,NO_2}$ following Eq. (12) at 100 % RH for snow covered ground. If $NO_2$ dry deposition persists into winter months at levels observed for late fall in Fig.

3, this represents a significant depositional sink for wintertime $NO_2$ not currently represented in CTMs when both the lifetime and near-surface concentrations of $NO_x$ are at a maximum.

### 3.3.4 Bottom-up estimates of nocturnal $V_d(NO_2)$

Simple estimates of bottom-up bulk-canopy $V_d(NO_2)$ provide a useful sanity check on top-down eddy covariance inferred values and are a starting point for a mechanistic explanation of bulk-canopy deposition. Bottom-up estimates of nocturnal

$R_c(NO_2)$ for Harvard Forest were computed from parallel contributions of uptake to leaves, bark, and the forest floor:

$$R_c(NO_2) = \left[ 1/r_{leaf} + 1/r_{bark} + 1/(r_a + r_{floor}) \right]^{-1} , \tag{13}$$

where in-canopy aerodynamic resistance $r_a$ was computed according to Zhang et al. (2003) as a prescribed land type specific value with LAI and friction velocity dependence (Table S3). Component canopy surface resistances in Eq. (13) were computed at hourly resolution following Eq. (11) using surface-specific $NO_2$ uptake coefficients from Table 1. Meteorological and

phenological data required to compute thermal speed $\bar{v}_t$ and surface area scale factor $\alpha$ in Eq. (11) are from observations (i.e., Fig. S4 & S5). Component surface area scale factors are material dependant, varying according to the surface area used to normalize deposition fluxes in corresponding measurement studies (Table 1 & S2). For complex surfaces with extensive and difficult to quantify substructures, planar (i.e., forest floor) or geometric (i.e., tree bark) surface areas are used. For more tractable surfaces such as deciduous or coniferous foliage, total leaf area or LAI are used. Attention must be paid to the

corresponding surface area for which a particular uptake coefficient is to be applied, and the understanding that application of



an uptake coefficient for a complex surface where planar or simple estimates of geometric surface areas were used during measurement (i.e., forest floor, snow, or bark) assumes surface area equivalence in subsequent applications. As seen in Table 1, nocturnal uptake of $NO_2$ to coniferous foliage was found to be 2.7 times greater than to deciduous leaves on a LAI basis under unsaturated (RH < 90 %) conditions. As is discussed further in Section 4, in addition to the greater total-to-projected

surface area of coniferous (~ 2.7) compared to deciduous (~ 2) leaves, we attribute the reduced LAI-weighted non-stomatal uptake of $NO_2$ to deciduous leaves as a consequence of the hydrophobic adaxial (top) surface of deciduous leaves which lack stomatal pores and therefore the elevated water vapour concentrations sufficient to support thin water films for reaction R1 to proceed under low to moderate ambient RH. The forest canopy is expected to be dew covered under conditions of high ambient RH > 96 % (Turnipseed et al., 2006), at which point we assume both top and bottom faces of deciduous leaves would support

thin water films; we therefore increase the $\alpha$ value used to scale uptake to deciduous leaves from LAI to 'total leaf area' (i.e., 2LAI) for RH > 96 %. As seen in Table 1, uptake to wet bark is twice that of dry bark, and we assume wetted bark for RH > 96 %. Also from Table 1, $NO_2$ uptake to snow is approximately one-third that of the forest floor; we make the assumption that the forest floor is snow covered in winter months (DJFM) for ambient temperatures < 0 °C.

The top panel of Fig. 4 depicts monthly mean estimates of nocturnal component canopy resistances used in Eq. (13)

to compute bottom-up bulk-canopy $R_c(NO_2)$. Also included are monthly mean estimates of top-down optimized bulk-canopy $r_{hyd}$ for both α values 1 and 2. The middle and bottom panels of Fig. 4 depict parameterized $V_d(NO_2)$ following Eq. (2) using both bottom-up (Eq. (13)) and top-down (Eq. (11)) estimates of bulk-canopy surface resistance. Also included are the fractional contributions that leaf, bark, and forest floor surfaces make to total canopy $NO_2$ uptake. Due to the compensating seasonal contributions of $r_{leaf}$ and $r_{floor}$ to total $NO_2$ deposition, bottom-up nocturnal $V_d(NO_2)$ shows little seasonality, in accordance with

eddy covariance observed $V_d(NO_2)$ from spring through fall (Fig. 3). As depicted in the middle panel of Fig. 4, bottom-up estimates of $V_d(NO_2)$ using uptake coefficients from Table 1 are greater than top-down optimized values across all seasons— a 27 % difference over the 12-month period when top-down $V_d(NO_2)$ is computed using α = 2, increasing to an 87 % difference for α = 1. Computing $NO_2$ uptake to dry and wet bark using the uptake coefficients from Table 1 of 5.0 x $10^{-6}$ and 1.0 x $10^{-5}$, respectively, results in bark surfaces being the predominant nocturnal dry depositional sink for $NO_2$ at Harvard Forest,

contributing about half the total nocturnal uptake across all seasons with the remaining half apportioned between the
forest floor (29 %) and foliage (21 %). Uptake coefficients to dry and wet bark in Table 1 are from Hanson et al. (1989), where chamber-measured $NO_2$ uptake was to Teflon end-capped cylindrical branch or trunk samples with diameters of ~ 15 cm. It is expected that the calculated cylindrical geometric surface area used to normalize uptake to exposed bark samples in Hanson et al. (1989) would be less than the surface area available for reaction due to bark roughness and shape complexity for samples

of this size. From Table 1, $NO_2$ uptake to smoother wood surfaces (i.e., wood board or plywood) at RH < 90 % is at least 3 times less than uptake to dry bark, which we speculate could result from a greater available surface area for the bark samples tested. The bark surface area for trunk and branch samples of ~ 15 cm in diameter for the species examined in Hanson et al. (1989) (shagbark hickory, tulip poplar, loblolly pine, and southern red oak) may be an overestimate of average bark surface area for the canopy at Harvard Forest. As depicted in the bottom panel of Fig. 4, reducing $NO_2$ uptake to bark by a factor of 2

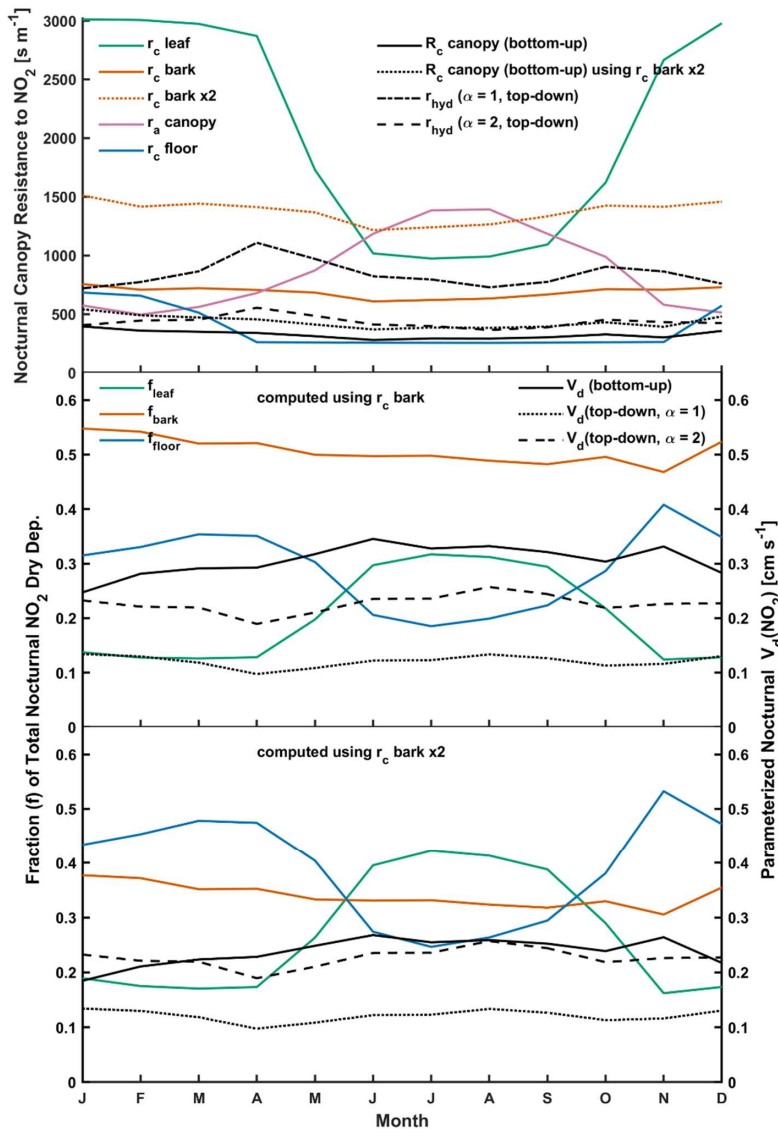

**Figure 4: (TOP)** Computed estimates of monthly mean component canopy resistances $r_c$ for nocturnal $NO_2$ uptake to leaves, bark, and the forest floor at Harvard Forest, and resulting 'bottom-up' bulk-canopy $R_c$. Also included are 'top-down' optimized bulk-canopy resistances $r_{hyd}$. **(MIDDLE)** Resulting 'bottom-up' and 'top-down' canopy-scale nocturnal deposition velocities $V_d(NO_2)$, including fractional contributions of leaf, bark, and forest floor surfaces to total canopy $NO_2$ uptake. **(BOTTOM)** As above (middle), however, with deposition fractions and bottom-up $V_d(NO_2)$ computed with uptake to bark reduced by a factor of two. Monthly values of resistances, deposition velocities, and meteorological inputs are included in Table S3 in the supplement. Surface-specific $NO_2$ uptake coefficients used to calculate bottom-up $r_c$ are included in Table 1.





(equivalent to a 2-fold increase in $r_c$ bark) results in bottom-up $V_d(NO_2)$ within the range of eddy covariance observed monthly values corrected for soil NO (0.2–0.3 cm s$^{-1}$) and within 2 % of top-down $V_d(NO_2)$ for α = 2 (parameterization P7) over a 12-month period. At this uptake level, bark is no longer the predominant sink for nocturnal dry deposition of $NO_2$, instead taking on a secondary role where predominant uptake alternates between canopy foliage during summer months and the forest floor during late fall, winter, and early spring. On an annual basis, nocturnal $NO_2$ uptake to forest surfaces are within 30 % of one another for the reduced bark uptake case; specifically, the forest floor accounts for 38 % of uptake, bark 34 %, and foliage 28 %. Bottom-up modelling estimates of canopy-scale dry deposition of $NO_2$ would benefit from future chamber studies detailing $NO_2$ uptake to a variety of bark surfaces over a range of humidities and temperatures.

**3.4 Evaluation of parameterized $V_d(NO_y)$ over Harvard Forest**

Simulated $V_d(NO_y)$ from base and updated parameterizations is evaluated against observations from the HFEMS, considering $NO_y$ component species $NO_2$, NO, $HNO_3$, and PAN. Of particular interest is the period from June–November 2000, when hourly observations of above-canopy $HNO_3$ concentration—a significant contributor to $NO_y$ dry deposition at this location (Horii et al., 2005)—was added to the suit of long-term measurements which include hourly concentrations of total $NO_y$ and component species NO, $NO_2$, and PAN alongside eddy covariance measurements of $NO_y$ flux (Fig. S7). The top panel of Fig. 5 depicts the diel climatology (June–November 2000–2002) of measured $NO_y$, NO, $NO_2$, $HNO_3$, and PAN over Harvard Forest. Also depicted is inferred $NO_y$ calculated from the sum of aforementioned component species. The middle panel of Fig. 5 depicts species-specific fractional contributions to measured $NO_y$. On average, inferred $NO_y$ from the sum of measured component species is ~ 76 % of measured total $NO_y$, with component species contributing 48 % ($NO_2$), 16 % ($HNO_3$), 8 % (PAN), and 4 % (NO). As discussed in Horii et al. (2005), Harvard Forest is influenced by two predominant airmasses: (i) northwesterly flow brining cool, dry, and less polluted air with an $NO_y$ concentration budget that is mostly closed by $NO_x$, $HNO_3$, and PAN and (ii) southwesterly flow consisting of warmer, humid, and significantly more polluted air wherein up to 50 % of the $NO_y$ budget remains unaccounted for, although the rank of measured contributions remains in the order $NO_2$ > $HNO_3$ > PAN > NO. The flux budget analysis of Horii et al. (2005) in their initial presentation of this dataset supported the presence of an unidentified rapidly depositing $NO_y$ species in southwesterly flows, corroborating suggestions that alkyl nitrates resulting from oxidation of biogenic isoprene and monoterpenes in the presence of $NO_x$ could contribute up to 25 % of summertime $NO_y$ deposition at this site (Munger et al., 1998).

To compare to $V_d(NO_y)$ inferred from measured fluxes, we compute simulated deposition velocities $V_{d,sim}(NO_y)$ from a linear combination of parameterized component deposition velocities $V_d(x_i)$ weighted by species-specific concentration fractions (Michou et al., 2005; Wu et al., 2011):

$$V_{d,sim}(NO_y) = \frac{\sum_i [x_i] V_d(x_i)}{\sum_i [x_i]}, \qquad (14)$$



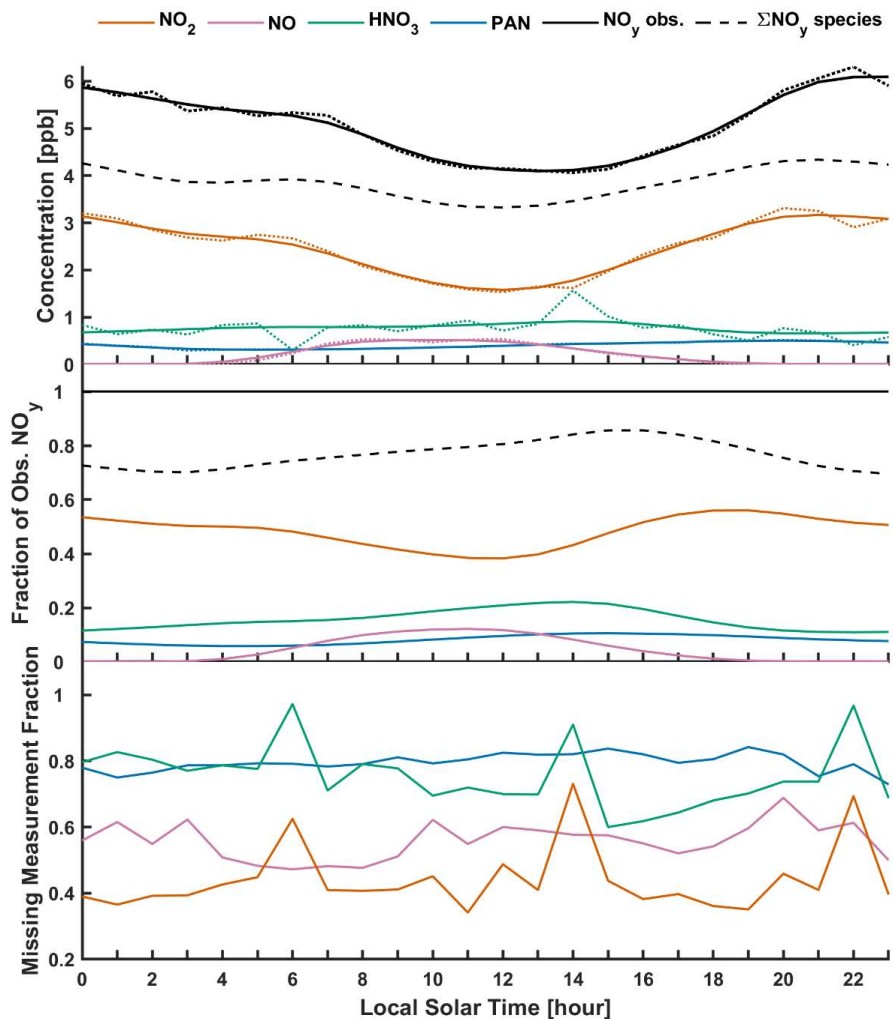

**Figure 5: (TOP)** Diel climatology of observed $NO_y$ and component species $NO_2$, $HNO_3$, PAN, and NO measured above Harvard Forest (June–November 2000–2002). Solid lines depict a smoothing spline fit to hourly mean concentrations (dotted). Also shown is the sum of smoothed $NO_y$ component species concentrations. **(MIDDLE)** Fractional contributions of $NO_y$ component species to observed $NO_y$. These values were computed as ratios of the smoothed diel mean concentrations in the top panel. Shown are individual $NO_y$ species fractions (colored as above) and their sum (dashed black). **(BOTTOM)** Fraction of hourly $NO_y$ component species concentrations (colored as above) missing from the measured $NO_y$ hourly time series spanning June–November 2000. As a gap-filling strategy in the calculated $V_d(NO_y)$ hourly time series, missing values were inferred using Eq. (15).





However, due to the large number of coincident hourly observations required for the comparison, only 19 coincident hourly values of $V_d(NO_y)$ and $V_{d,sim}(NO_y)$ exist across the entire data set consisting of over 2000 hourly measurements of $V_d(NO_y)$ from June–November 2000. For this reason, a gap-filling method is employed to estimate date ($d$) and hour ($h$) specific missing NO$_y$ component concentrations $[x_i]_{d,h}^{infer}$ for NO$_2$, NO, HNO$_3$, and PAN from measured NO$_y$:

$$[x_i]_{d,h}^{infer} = \left( \frac{\overline{[x_i]}_h^{meas}}{\overline{[NO_y]}_h^{meas}} \right)_{clim} [NO_y]_{d,h}^{meas}, \tag{15}$$

where the diel climatologies of component fractions $\overline{[x_i]}_h^{meas}/\overline{[NO_y]}_h^{meas}$, depicted in the middle panel of Fig. 5, are computed over June–November 2000–2002 and subjected to a smoothing spline fit. This method of gap-filling was employed by Wu et al. (2011) in their application of this dataset to evaluate simulated NO$_y$ deposition velocities from the WRF-Chem and NOAH-GEM dry deposition modules—a difference herein being that we compute component fractions as the 'ratio of smoothed means' rather than the 'mean of ratios' to reduce the effect of outliers (data not shown). The fraction of inferred species-specific hourly concentrations required for gap-gilling over the study period is depicted in the bottom panel of Fig. 5 as the missing measurement fraction. A large fraction of inferred values for PAN results from the absence of observations from August–November 2000, thus relying on years 2001–2002 to inform the climatology using Eq. (15) (Fig. S7). HNO$_3$ was also inferred to a large degree; although hourly concentrations were measured fairly consistently from June–November 2000 (Fig. S7), monthly coverage was only ~ 20 % (Fig. 5). Prior to discussing model–measurement comparison of diel mean $V_d(NO_y)$ in Section 3.4.2, we first discuss the simulated diel profiles of $V_d(x)$ used in Eq. (14) to compute simulated $V_d(NO_y)$.

**3.4.1 Simulated diel profiles of $V_d(x)$ for measured NO$_y$ component species**

Figure 6 depicts simulated diel mean deposition velocities from selected updated parameterizations for NO$_y$ component species HNO$_3$, PAN, and NO$_2$ over Harvard Forest, aggregated from hourly values computed using observed meteorological and phenological inputs. Corresponding simulated component resistances $R_a$, $R_b$, and $R_c$ are depicted in Fig. S6. Unless otherwise indicated, aerodynamic resistance was computed from the measurement height of 29 m. Depicted simulations of $V_d(HNO_3)$ include parameterizations P2 (equivalent to P1 referenced from measurement height; Table 2), P3 (RSL correction assuming $u(z_o) > 0$ m s$^{-1}$), P5 (improved calculation of molecular diffusivity), and P8 referenced from both the measurement height (29 m) and the center of GEOS-Chem's lowest grid box (~ 60 m). The computation of $V_d(HNO_3)$ between parameterizations P5 and P8 is equivalent, i.e., identical formulations of $R_a$ and $R_b$ (Table 2). The small increase in daytime $R_a$ of ~ 15 % due to the incorporation of the RSL in parameterization P3 (Fig. S6) results in a small (~ 7 %), yet significant (p < 0.05), decrease in daytime $V_d(HNO_3)$ compared to P2 values—a slightly greater change than observed over Talladega National Forest (Section 3.1, Table 3) where a higher relative measurement height (2 $h_c$ vs. 1.5 $h_c$ at Harvard Forest) dampened the effect of the RSL on $R_a$ computed from this altitude. Similarly, the lower depth of influence of the RSL during nocturnal conditions results in P3 updates having a reduced effect on nighttime $V_d(HNO_3)$. Due to low aqueous solubility of NO$_2$ and PAN, $R_c$ is the dominant

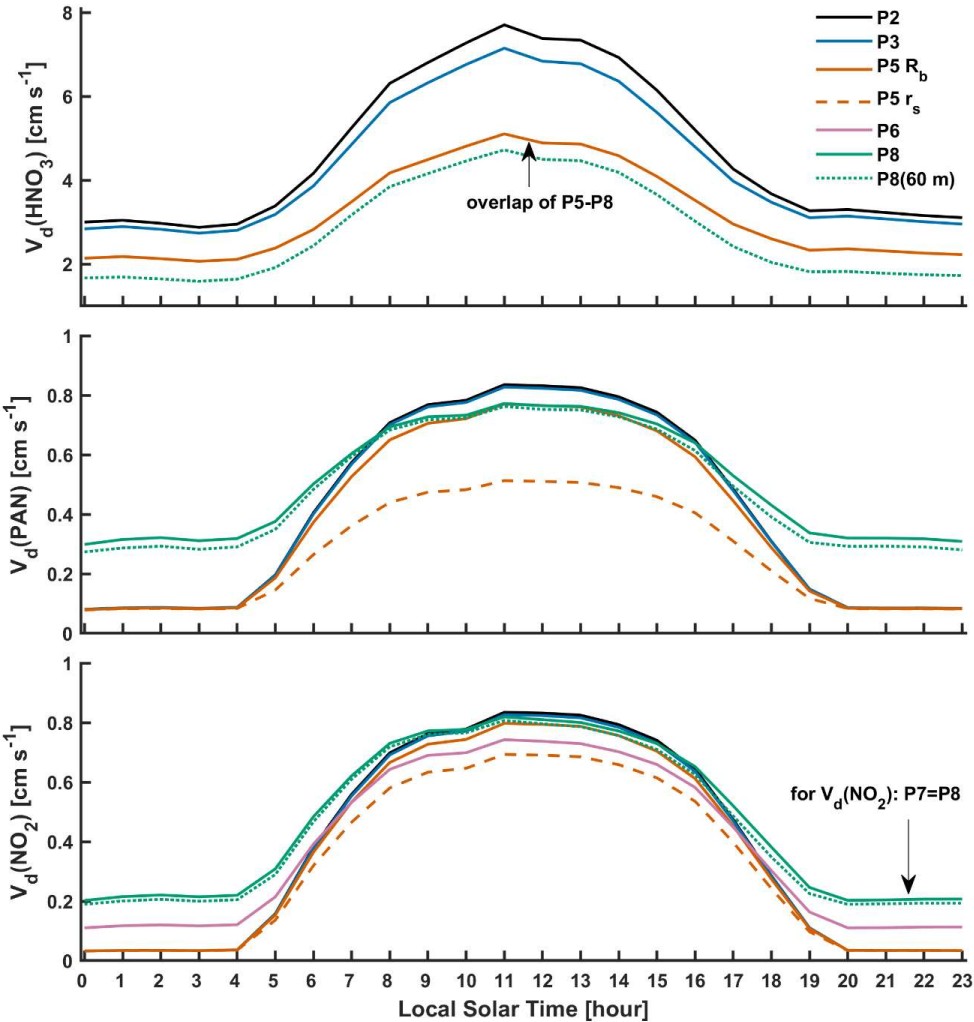

**Figure 6:** Simulated diel mean deposition velocities for HNO$_3$, PAN, and NO$_2$ over Harvard Forest (June–November). In addition to depicted parameterizations from Table 2, shown for parameterization P5 is the cumulative effect of molecular diffusivity updates to quasi-laminar sublayer resistance $R_b$ followed by resistance to stomatal uptake $r_s$, observable for daytime PAN and NO$_2$ which deposit under stomatal control. Simulated deposition velocities were computed from the measurement reference height of 29 m, unless otherwise indicated (i.e., P8(60 m)). Diel mean values are from a continuous hourly dataset computed using observed meteorological and phenological (LAI, canopy height) inputs. Component resistances $R_a$, $R_b$, and $R_c$ are shown in Fig. S6 in the supplement.



term in the resistance pathway for these species (Fig. S6) outside of infrequent very stable conditions (Fig. S1); accordingly, RSL corrections to $R_a$ in parameterization P3 have negligible influence on resulting deposition velocities for these species. Large reductions in simulated $V_d(HNO_3)$ are seen for parameterization P5, where the use of accurate molecular diffusivities results in an increase in $R_b(HNO_3)$ of ~ 95 %. Associated increases in $R_b(NO_2)$ and $R_b(PAN)$ of ~ 60 % and 110 %, respectively,

result in insignificant reductions to $V_d(NO_2)$ across all times of day and small reductions in daytime $V_d(PAN)$ of 7 % due to the dominant contributions of $R_c$ for these species. However, due to the dependence of species-specific stomatal conductance on the ratio of molecular diffusivities $D_x/D_{H_2O}$, diffusivity updates to parameterization P5 result in increased stomatal resistances with notable reductions in daytime dry deposition for species that deposit under stomatal control—up to 13 % and 32 % for $NO_2$ and PAN, respectively. At night when stomates are assumed to be closed ($r_s > 10^4$ s m$^{-1}$), non-stomatal branches of $R_c$

control deposition, therefore reducing the effects of updates to molecular diffusivity on the nocturnal dry deposition of $NO_2$ and PAN as depicted in Fig. 6.

Included in Fig. 6 for simulated $V_d(NO_2)$ is the effect of replacing the non-stomatal branch of $R_c$ with r$_{hyd}$ according to Eq. (11), resulting in large increases in nocturnal $V_d(NO_2)$ of up to a factor of 6, as depicted in parameterizations P6 & P8. The relative increase in daytime $V_d(NO_2)$ is much less (24% for P8 with α = 2) due to competing stomatal uptake, however,

enough to restore peak daytime $V_d(NO_2)$ to base levels. The reduced diurnal variability in simulated $V_d(NO_2)$ seen for parameterization P8, ~ 4-fold compared to 20-fold for P2, is consistent with the diel cycles in $V_d(NO_2)$ inferred from canopy-scale observations where daytime values are on the order of 2 to 7 times greater than at night (Eugster and Hesterberg, 1996; Hanson and Linderg, 1991; Plake et al., 2015; Rondón et al., 1993; Stella et al., 2013; Walton et al., 1997). Greater diel variation in $V_d(NO_2)$ is seen in leaf-level uptake studies, where daytime deposition velocities are on average an order of

magnitude greater than in the absence of photosynthetically active radiation (Delaria et al., 2020, 2018).

Turnipseed et al. (2006) present eddy covariance flux observations of PAN over a summertime coniferous forest in North Carolina, finding appreciable nocturnal dry deposition that increases when the canopy is wet—well-above predicted values from the W89 parameterization. Accordingly, parameterization P8 includes suggested empirical updates for dry deposition of PAN developed by Turnipseed et al. (2006) for forested ecosystems, namely, setting non-stomatal resistance to

cuticular deposition ($r_{lu}$ in Eq. (5)) to 250 s m$^{-1}$ for dry foliage and 125 s m$^{-1}$ for wet foliage. Turnipseed et al. (2006) define leaf surfaces as wet during and immediately following precipitation events or when above-canopy RH > 96 %; herein, we define the canopy as wet when above-canopy RH > 96 %. To extend applicability to other forest locations, we scale recommended cuticular resistances by the ratio LAI$_{HFEMS}$ / 3.5, where 3.5 m$^2$ m$^{-2}$ was the LAI at the study site of Turnipseed et al. (2006). This update to non-stomatal uptake of PAN in parameterization P8 reduces median nocturnal $R_c(PAN)$ over

Harvard Forest from ~ 1000 s m$^{-1}$ to 200 s m$^{-1}$ (Fig. S6), resulting in nocturnal and daytime increases to $V_d(PAN)$ of 250 % and 60 %, respectively (Fig. 6). As was seen with updates to $V_d(NO_2)$ in Fig. 6, parameterization P8 updates to $V_d(PAN)$ largely restore reduced daytime values in parameterization P5 to P2 base levels. It is noted that the empirical update from Turnipseed et al. (2006) is not mechanistically based, nor is it clear as to the general applicability to other land types, locations, or seasons. As is often the case in parameterizations of dry deposition processes, further study is warranted. As will be discussed for $NO_2$



in Section 4, recent chamber studies of foliar uptake of PAN both question (Place et al., 2020) and support (Sun et al., 2016) the role of non-stomatal deposition, rendering dry deposition of PAN an ongoing active area of research.

Studies comparing CTM-simulated deposition velocities to measurement-inferred values often reference $R_a$ from CTM grid-box-center instead of measurement heights (Clifton et al., 2017; Nguyen et al., 2015; Nowlan et al., 2014; Silva and Heald, 2018). Increases in $R_a$ when referenced from the center of GEOS-Chem's lowest level ($\sim 60$ m) instead of the 29 m
measurement height over Harvard Forest results in moderate, although significant ($p < 0.05$), reductions in simulated $V_d(HNO_3)$ of 10 % (daytime) to 20 % (nighttime), as depicted in Fig. 6 by comparing parameterization P8 with P8(60m). These moderate increases in $R_a$ are insufficient to cause significant change to either $V_d(PAN)$ or $V_d(NO_2)$ which deposit under $R_c$ control.

**3.4.2 Measurement–model comparison of $V_d(NO_y)$**

Observations of hourly above-canopy eddy covariance fluxes of $NO_y$ at the HFEMS are mostly downward (> 99 %, Fig. S8),
regardless of adjustment for soil-emitted NO. Figure 7 depicts observation-inferred diel mean $V_d(NO_y)$ alongside simulated values from parameterizations P2, P5, and P8—selected from Table 2 to highlight the dominant effects of diffusivity updates (P5) and surface resistance updates to $NO_2$ and PAN (P8) on simulated $V_d(NO_y)$ referenced from the measurement height at Harvard Forest (P2). Inferred $V_d(NO_y)$ was calculated from eddy covariance observed $V_{ex}(NO_y)$ adjusted for estimates of soil-emitted NO, analogous to $V_d(NO_2)$ in Eq. (9). As evident from Fig. 7, soil NO corrections to observed $V_{ex}(NO_y)$ result in small
increases to inferred $V_d(NO_y)$—9 % for parameterizations P2 and P5, decreasing to 7 % for P8 due to an increase in the simulated canopy reduction factor (Fig. S3) from updates to non-stomatal $NO_2$ uptake. By far the largest contributor to simulated $V_d(NO_y)$ is $HNO_3$ (Fig. 7), contributing over 75 % to 24-hour $NO_y$ flux for parameterization P2 (Table 5). Despite nocturnal mean $NO_2$ concentrations on the order of 3 ppb (Fig. 5), $NO_2$ makes near-negligible contributions to simulated nocturnal $V_d(NO_y)$ in parameterizations P2 and P5. Updates to the parameterization of molecular diffusivity (P5) results in
large reductions in simulated $V_d(NO_y)$ (-30 % day; -28 % night) and resulting 24-hour depositional flux (26 % reduction, Table 5) due to large reductions in simulated $V_d(HNO_3)$ (Fig. 6). This also exposes a morning peak in inferred $V_d(NO_y)$ (Fig. 7) which simulated values fail to capture. By monitoring the rate of change of vertically integrated in-canopy concentration profiles, Munger et al. (1996) showed that canopy storage contributions to above-canopy $NO_x$ and $O_3$ fluxes were small at Harvard Forest; however, as noted by Horii et al. (2005), $NO_y$ fluxes measured at the HFEMS did not include a storage term as canopy
vertical profiles of $NO_y$ were not available. Geddes et al. (2014) measured eddy covariance fluxes of $NO_y$, NO, and $NO_2$ above two midlatitude mixed hardwood forests, noting problematic interpretation of $NO_y$ fluxes between the hours of 07:30–11:00 LST due to suspected canopy storage contributions. Due to the absence of in-canopy $NO_y$ measurements from which to evaluate canopy storage, Geddes et al. (2014) considered observations of above-canopy $NO_y$ flux unrepresentative of the depositional flux over this timeframe and excluded observations during these hours from analysis. The anomalous morning
peak in observation-inferred $V_d(NO_y)$ in Fig. 7 could, in part, be due to canopy storage contributions at Harvard Forest. It is conceivable that mixing down of rapidly depositing $NO_y$ species (i.e., $HNO_3$ or organic nitrates), or gas-particle partitioning



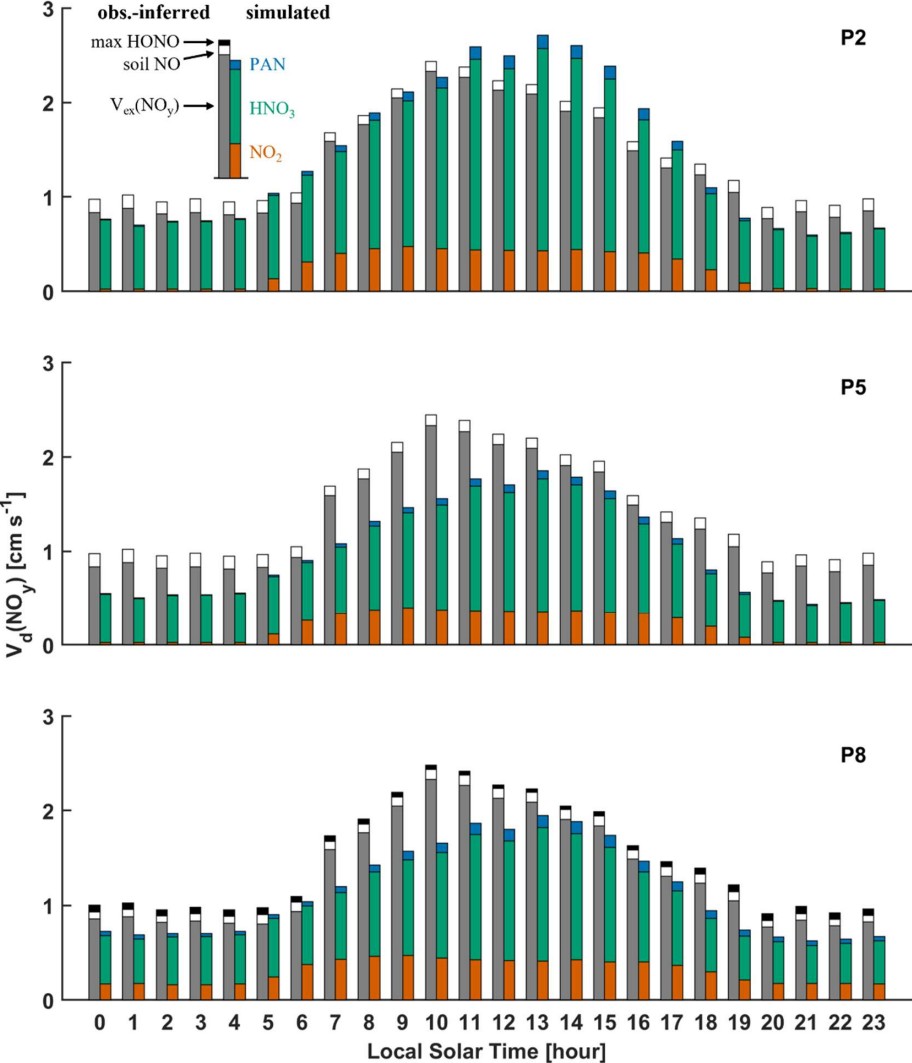

**Figure 7:** Simulated and observation-inferred diel mean $NO_y$ deposition velocities over Harvard Forest (June–November). In addition to observed $V_{ex}(NO_y)$, inferred contributions to $V_d(NO_y)$ from estimated soil NO emissions (Eq. (10)) and maximum HONO emitted following reaction R1 are also depicted. Simulated $V_d(NO_y)$ is depicted as the weighted sum (Eq. (14)) of contributing $NO_y$ component species $NO_2$, $HNO_3$, and PAN for three simulation types: **P2** (base), **P5** (updated $R_a$ and diffusivity $D$), and **P8** (updated $R_c(NO_2)$ and $R_c(PAN)$).



of ammonium nitrate, as growth of the morning boundary layer erodes the residual layer above could induce a downward spike in above-canopy $NO_y$ flux. It is also possible that a morning low bias in simulated $V_d(NO_y)$ results from heavy reliance on the climatological diel profile of $HNO_3$ (Fig. 5) due to the paucity of hourly $HNO_3$ observations (Fig. S7). Future analysis of this

measurement–model discrepancy would benefit from in-canopy vertical profiles of $NO_y$, particulate speciation, and component species of both at hourly resolution. As seen for parameterization P8 in Fig. 7, updates to non-stomatal deposition of $NO_2$ and PAN result in noticeable increases in simulated $V_d(NO_y)$ (16 % day; 30 % night), with updates to $NO_2$ deposition having a greater relative impact at night (25 % increase in $V_d(NO_y)$) than during the day(5 % increase in $V_d(NO_y)$). Parameterization P8 updates are associated with an 18 % increase in inferred 24-hour $NO_y$ flux due to large increases in inferred dry deposition of

$NO_2$ (56 % increase) and PAN (85 % increase) (Table 5).

**Table 5:** Inferred mean fluxes of $NO_y$ and component species over Harvard Forest.

| Parameterization[a] | Flux [ngN m$^{-2}$ s$^{-1}$] [b] | | | |
|---|---|---|---|---|
| | $HNO_3$ | $NO_2$ | PAN | $NO_y$[c] |
| P2 (base) | $19 \pm 24$ | $4.4 \pm 7.1$ | $1.1 \pm 1.4$ | $25 \pm 30$ |
| P3 ($R_a$ update) | $18 \pm 22$ | $4.3 \pm 7.0$ | $1.1 \pm 1.5$ | $23 \pm 29$ |
| P5 ($D$ update) | $13 \pm 16$ | $3.7 \pm 5.9$ | $0.7 \pm 0.9$ | $17 \pm 21$ |
| P7 ($R_c(NO_2)$, $\alpha = 2$) | $13 \pm 16$ | $5.8 \pm 7.4$ | $0.7 \pm 0.9$ | $19 \pm 22$ |
| P8 ($R_c(PAN)$) | $13 \pm 16$ | $5.8 \pm 7.4$ | $1.3 \pm 1.4$ | $20 \pm 23$ |

[a] See Table 2 for serial parameterization updates. Briefly,
   P2: equivalent to base P1 referenced from Harvard Forest measurement height of 29 m.
   P3: updates to aerodynamic resistance ($R_a$).
P5: updates to molecular diffusivity ($D$).
   P7: update resistance to surface uptake of $NO_2$ ($R_c(NO_2)$) to include reaction R1 using surface area scale factor $\alpha = 2$ (Eq.(11)).
   P8: update resistance to surface uptake of PAN ($R_c(PAN)$).
[b] 24 h mean ($\pm 1\sigma$) fluxes inferred from the product of simulated $V_d(x)$ and gap-filled measured concentrations over June–November 2000.
[c] Inferred $NO_y$ flux from the sum of inferred component ($HNO_3$, $NO_2$, and PAN) fluxes.


Also depicted in Fig. 7 is the effect on observation-inferred $V_d(NO_y)$ from a maximum estimate of HONO emitted from the heterogeneous hydrolysis of $NO_2$ on deposition surfaces following reaction R1. Assuming unity for HONO surface emission to the gas phase and subsequent ventilation from the canopy produces a 4 % increase in inferred $V_d(NO_y)$. Uncertainties exist around the nature of the dynamic equilibrium that establishes between evolved and adsorbed/dissolved

HONO (Collins et al., 2018; Harrison et al., 1996; Lee, 2012; Spicer et al., 1993; Wojtal et al., 2011) and subsequent implications of a nocturnal reservoir of deposited HONO as a daytime source of HONO to the atmospheric surface layer (He et al., 2006; Ren et al., 2020; VandenBoer et al., 2014, 2015; Wentworth et al., 2016). Lee et al. (2012) monitor near continuous above-canopy $NO_2$ and HONO concentrations and eddy covariance fluxes at the HFEMS during 2011, finding nocturnal enhancements in HONO concomitant with $NO_2$. However, neither upward nor downward fluxes of HONO were observed,

suggestive of establishment of dynamic equilibrium between HONO emission and deposition at Harvard Forest, where



perturbation fluxes were below detection limits. Measurements of HONO and $NO_2$ fluxes over grassland and sugar beet surfaces have highlighted the bidirectional nature of HONO exchange; HONO emission was found to dominate the bidirectional flux under elevated $NO_2$ concentrations ( >10 ppb for the land types studied) while deposition was noted to be dominant at lower ambient $NO_2$ concentrations (Harrison et al., 1996; Harrison and Kitto, 1994). The diel flux behaviour of

HONO is likely multifactorial, depending on land type, meteorology, and trace gas and particulate concentrations (Pusede et al., 2015; VandenBoer et al., 2015). The absence of significant fluxes of HONO over Harvard Forest as noted by Lee et al. (2012), despite observed downward nocturnal fluxes of $NO_2$ and nocturnal enhancement of HONO, does not exclude reaction R1 as a source of HONO. Rather, this indicates the importance of deposition and re-emission processes from canopy surfaces which may dominate HONO behaviour at rural forest sites (Ren et al., 2011, 2020; Sörgel et al., 2011; Zhou et al., 2002) in

the absence of strong pulses of ambient $NO_2$ perturbing the dynamic equilibrium between adsorbed/dissolved and gas phase HONO—as is routinely observed in laboratory studies (Finlayson-Pitts et al., 2003; Spicer et al., 1993).

Neglecting the possible effect of emitted HONO on $V_d(NO_y)$ given the findings of Lee et al. (2012), we find a moderate underestimate of 20 % in simulated $V_d(NO_y)$ (24-hour) from parameterization P8, with similar daytime and nighttime biases of -20 % and -19 %, respectively; excluding the period from 07:00–11:00 reduces the daytime bias to -15 % and 24-hour bias

to -17 %. As previously mentioned, the $NO_y$ concentration budget at the HFEMS is closed to 76 % on average from observations of $NO_x$, $HNO_3$, and PAN. Horii et al. (2005) provide evidence of a rapidly depositing unidentified $NO_y$ species at this site, especially under southwesterly flow, and speculate the unidentified $NO_y$ species as organic nitrates.

**4 Assumption of nocturnal stomatal closure**

Despite longstanding uncertainty regarding nocturnal stomatal behaviour (Caird et al., 2007; Costa et al., 2015; Dawson et al.,

2007), it is generally assumed that at night in response to elevated guard cell $CO_2$ concentration the stomata of C3 and C4 plants are nearly closed (Nobel, 2009), thereby shunting trace gas exchange. In both the widely used W89 and Z03 dry deposition schemes, stomata are assumed to be closed at night and therefore all nocturnal deposition parameterized through non-stomatal pathways. Although chamber studies consistently find stomatal control dominates daytime foliar uptake of $NO_2$, uncertainty remains regarding the importance of non-stomatal pathways at night. Some studies note negligible non-stomatal

contributions to $NO_2$ deposition, attributing nocturnal uptake to partially open stomata (Chaparro-Suarez et al., 2011; Delaria et al., 2020; Gebler et al., 2000; Rondón et al., 1993; Sparks et al., 2001), while others find non-stomatal contributions to be non-negligible (Geßler et al., 2002; Hanson et al., 1989; Thoene et al., 1996; Wang et al., 2020; Weber and Renenberg, 1996). This amalgam of contradictory results warrants further consideration and is discussed in the context of a literature review in Section S4 of the supplement.

Notwithstanding nocturnal stomatal behaviour, nocturnal ambient humidity is often elevated at locations with lush vegetation, as it is for summer months at Harvard Forest (monthly median RH 88–91 % at 15 m, Table S3), whereby growth of aqueous films have been observed to occlude stomatal pores (Grantz et al., 2018); heterogeneous hydrolysis of $NO_2$ would



be expected to proceed on foliar as well as other surfaces under these conditions. Thoene et al. (1996) monitored $NO_2$ uptake to Norway Spruce (*Picea abies*) using a branch enclosure, proposing thin water films forming on needle surfaces as a plausible

explanation for the observed correlation of $NO_2$ uptake to RH—a correlation which could not be explained by changes in stomatal conductance. Furthermore, nocturnal uptake of $NO_2$ through a deposition process involving HONO production has been implicated in the field on several occasions (Harrison and Kitto, 1994; Ren et al., 2020; Stutz et al., 2002), including with RH dependence (Stutz et al., 2004; VandenBoer et al., 2013).

   Inferred values of $\gamma_{NO_2}$ in Tables 1 & S2 fall within the range expected for uptake due to $NO_2$ heterogenous

hydrolysis, i.e., $10^{-6}$ to $10^{-5}$ (Section S2). For foliar surfaces under dark conditions, 'total leaf area' normalized uptake was not observed to exceed that of a planar surface of distilled water, supporting the possibility that reaction R1 is the predominant mechanism driving uptake to both the non-foliar and foliar surfaces presented. Three features stand out from tabulated values of $\gamma_{NO_2}$ in Tables 1 and S2. First, a dependence on surface moisture—dependence on RH or surface wetness is seen for uptake to wood board, concrete, and tree bark. The RH or wetness dependence for $NO_2$ uptake to wood board and bark is similar to

the factor of two increase in $\gamma_{g,NO_2}$ from Eq. (12) between RH 50% and 100%. Second, surface area available for heterogeneous reaction has direct influence on resulting material-specific uptake, as expected for a collision-limited heterogeneous process such as reaction R1. Surfaces with complex and undetermined microscopic surface areas (i.e., bark, coarse concrete, forest floors, and snow) exhibit higher $V_d^{surf}$ and resulting $\gamma_{NO_2}$— a factor of 2 to 30 greater than to surfaces normalized by accurate predictions of available surface area (i.e., bulk water and foliar). This increased uptake to convoluted

surfaces could be an indirect measure of the total microscopic surface area available for reaction, thus highlighting the utility of using field-derived uptake coefficients for parameterizing surface uptake in dry deposition models. Third, a feature stands out between $NO_2$ uptake to coniferous versus deciduous leaves when stomatal aperture is at a minimum. When uptake is normalized by LAI, inferred $\gamma_{NO_2}$ to coniferous species is on average 2.7 times greater than to deciduous species—a factor equal to the ratio of total needle surface area-to-LAI (Riederer et al., 1988). Since coniferous needles have stomata distributed

across the entire leaf surface, whereas most deciduous leaves have stomata located on the lower (abaxial) leaf surface and a thicker hydrophobic wax cuticle on the upper (adaxial) leaf surface, the 2.7-fold greater uptake to coniferous species (when normalized by LAI) may reflect the absence of thin water films on the adaxial surface of deciduous leaves . These inferences are consistent with the work of Summer et al. (2004), where similar rates of $NO_2$ heterogeneous hydrolysis across a variety of hydrophilic and hydrophobic surfaces were understood in the context of available surface areas supporting thin water films.

**5 Conclusions**

Extraction of the trace gas dry deposition algorithm from GEOS-Chem and reimplementation to run offline in single-point-mode enabled a detailed and more direct evaluation of various branches of the algorithm against eddy covariance observed deposition velocities over two North American temperate forest ecosystems. Observations of deposition velocities for species



that deposit under dynamical control (i.e., nominally small resistance to surface uptake $R_c$) facilitated the identification of a large high bias in computed molecular diffusivities. Correction of this bias using Fuller's method to calculate diffusivities in the absence of measured values resulted in improved simulation of $V_d(HNO_3)$, and consequently, improved representation of simulated $V_d(NO_y)$ to an extensive dataset spanning many months wherein a potentially anomalous morning peak in observed $V_d(NO_y)$ was exposed, in agreement with previous work demonstrating problematic interpretation of late morning $NO_y$ fluxes over a similar forest ecosystem (Geddes and Murphy, 2014). Site-specific roughness length and reference height were found to be important constraints on the calculation of aerodynamic resistance for rapidly depositing species, whereas correction for the influence of the roughness sublayer was found to be of minor importance at the measurement heights involved and of negligible effect at the dry deposition reference height used in GEOS-Chem, in agreement with previous work (Simpson et al., 1998).

A large low bias in simulated nocturnal $V_d(NO_2)$ against eddy covariance observed values spanning many months over Harvard Forest was identified in accordance with previous work (Horii, 2002) which included non-stomatal $NO_2$ dry deposition simulated following the W89 algorithm (Wesely, 1989)—a scheme known to misrepresent non-stomatal surface uptake for off-target species such as $NO_2$ (Zhang et al., 2003b). We addressed this low (-80 %) bias by representing $NO_2$ heterogeneous hydrolysis (reaction R1)—a well-known surface reaction (Finlayson-Pitts et al., 2003) shown to be of importance in the field (VandenBoer et al., 2013), however, yet to be represented in dry deposition parameterizations to our knowledge—in the calculation of non-stomatal surface resistance. A literature review of surface-specific $NO_2$ deposition velocities to both non-foliar and nocturnal foliar surfaces highlights the importance of considering microscopic surface area for heterogeneous reaction, and enabled estimates of bottom-up $V_d(NO_2)$ for Harvard Forest which agree well with top-down estimates optimized from eddy covariance observed values when uptake to bark was reduced by a factor of two, indicating a need for further laboratory study of uptake to bark for representative samples. Consideration of soil NO emission on eddy covariance observed $V_d(NO_2)$ at Harvard Forest was found to be important, as was representative canopy surface area when applying $NO_2$ uptake coefficients from field and laboratory observations.

We persist with the assumption that nocturnal uptake of $NO_2$ follows non-stomatal pathways, as is currently the case in dry deposition schemes widely used in atmospheric CTMs. Meanwhile, the nocturnal behaviour of stomata remains an active area of research. Confounding processes such as the hydraulic activation of stomata (HAS) and condensation at elevated RH complicate the inference of stomatal conductance to trace gases from observations of water vapor flux, especially in well mixed chambers under dark conditions. It would be helpful for future enclosure studies of $NO_2$ uptake to consider the effects of heterogeneous hydrolysis of $NO_2$ on foliar and non-foliar surfaces, as well as potential biases in estimates of stomatal conductance resulting from possible HAS.

We recommend the implementation of a mechanistic non-stomatal dry deposition scheme for $NO_2$ in atmospheric models that considers the effect of reaction R1 on surface resistances. This represents a significant depositional sink for $NO_2$ under conditions when both the lifetime and near-surface concentration of $NO_2$ may be elevated, i.e., nocturnal and urban wintertime conditions. We present two approaches that result in general agreement for a mature temperate forest ecosystem.



The simplest approach being to represent non-stomatal resistance to $NO_2$ uptake as $r_{hyd}$ following Eq. (11) with $\alpha = 2$ for high surface area land types such as urban and forest, and $\alpha = 1$ for remaining land types. Long-term field studies quantifying atmosphere–surface exchange across a variety of land types and seasons would facilitate further development of species-specific dry deposition pathways.

Although a main focus of this work was on the effect that reaction R1 had on $V_d(NO_2)$, there is much interest in an accurate HONO simulation given that the near-surface nocturnal build-up of HONO results in an early morning burst of OH and NO radicals as HONO photolyzes (Finlayson-Pitts, 2009; Ren et al., 2020), initiating photochemistry prior to other $HO_x$ precursors (Platt et al., 1980). Future work will present the implementation of updates to $V_d(NO_2)$ developed herein into the GEOS-Chem CTM, including analysis of the resulting HONO simulation.

*Code and data availability*. The source code for the dry deposition module from GEOS-Chem version 10-01 is available in a Zenodo repository (https://zenodo.org/records/13892258) (Boys, 2024). Harvard Forest data used herein, including trace gas concentrations and fluxes (Munger and Wofsy, 2004, 2023), meteorological observations (Fitzjarrald and Sakai, 2023; Munger and Wofsy, 2024), and LAI (Matthes et al., 2024) are publicly available from the Harvard Forest Data Archive (https://harvardforest.fas.harvard.edu/data-archive).

*Author contributions.* The manuscript was written using contributions from all authors. BLB conceived and designed the study under the supervision and support of RVM. BLB performed the model simulations and data analysis with feedback from RVM. BLB wrote the original draft with feedback from RVM. TCV provided an extensive review, edit, and direction of the original draft which was incorporated into subsequent drafts. All authors have reviewed, edited, and approved the final version of the manuscript.

*Competing interests.* No competing interests are present.

**Acknowledgements.** This work was supported by the Natural Sciences and Engineering Research Council of Canada. BLB acknowledges support from an Izaak Walton Killiam Memorial Scholarship. RVM acknowledges support from the U.S. National Science Foundation Grant 2244984. We gratefully acknowledge the publicly available Harvard Forest dataset maintained by Drs. Steve Wofsy and Bill Munger that enabled much of this work, and many helpful discussions with Dr. Glen Lesins.



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
