# Peer review of "Evaluation and updates to the oxidized reactive nitrogen trace gas dry deposition parameterization from the GEOS-Chem CTM, including a pathway for ground surface NO2 hydrolysis"

_EGUsphere, 2024_

## Author Response (AR2)

We would like to thank both Reviewers for their thoughtful and constructive feedback and acknowledge that their efforts have helped to improve the utility and focus of the revised manuscript. We summarize the main changes made in the revised manuscript below, followed by point-by-point responses (blue text) to Reviewer Comments, RC1 and RC2 (black text). Specific changes in manuscript (*blue italicized*) are preceded with section/figure and/or line numbers (**bold**) that refer to the revised manuscript (without tracked changes).

**Summary of main changes in revised manuscript**

The revised manuscript has been updated in two main ways. First, following the recommendations of Reviewers 1 & 2, to improve the focus of the manuscript by avoiding lengthy and overly detailed descriptions that distract from the main points. The revised manuscript was edited with this in mind and ~5 pages worth of text has been removed/moved to supplement. Second, following the request of Reviewer 1, we implemented the Zhang 2003 (Z03) dry deposition scheme to discuss comparison with updates made to non-stomatal $NO_2$ dry deposition. Since the Z03 scheme is a development over the Wesely 1989 (W89) dry deposition scheme used by GEOS-Chem, this request was well-justified and worth the effort to address.

**RC1**

This study conducted extensive sensitivity tests on the dry deposition code of GEOS-Chem focusing on comparing modeled NO2 dry deposition velocity (Vd) with eddy covariance measurements collected at the Harvard Forest station. The method used in this study is scientifically sound. Results from these tests are properly interpreted and may help understand the causes of model-measurement biases and reduce model uncertainties. One addition from this study to the existing literature is the modification of the GEOS-Chem deposition scheme by including the effect of NO2 hydrolysis at surface so that nocturnal NO2 Vd is increased, which is worth to be published, although its general application may need further evaluation in future studies.

Response: We agree that this work is by no means a final word. It is our hope that it prompts future developments and evaluation. This was a first attempt to implement a long-known surface

reaction for $NO_2$ in the context of a tractable dry deposition scheme for use in a CTM. To better convey this point, in the revised manuscript we have:

(1) removed strong language claiming our updated parameterization to be unbiased:

**(changed in Abstract)** (i) *"We eliminate a large low bias..."* **changed to (L15)** *"We address a large low bias..."* and (ii) *"These developments are applicable to models across scales, having important implications..."* **changed to (L21)** *"These developments are a first step towards a tractable representation of $NO_2$ hydrolysis in a dry deposition scheme and have important implications..."*

**(changed in Section 3.3.1)** *"Mechanistic updates to the parameterization of $NO_2$ surface uptake (P6–P7) developed to remedy large biases in nocturnal $V_d(NO_2)$ computed following the W89 scheme (P5) are discussed in Section 3.3.3."* **changed to (L463)** *"In Section 3.3.3, we evaluate a simple representation of non-stomatal $NO_2$ uptake following reaction R1 against eddy covariance inferred $V_d(NO_2)$."*

**(changed in Section 3.3.3)** *"Increasing the rate of non-stomatal uptake of $NO_2$ by computing $r_{hyd}$ with $\alpha = 2$ in parameterization P7 resulted in an unbiased simulation of nocturnal mean $V_d(NO_2)$ of $0.24 \pm 0.04$ cm s$^{-1}$ ... "* **changed to (L559)** *"Increasing the rate of non-stomatal uptake of $NO_2$ by computing $r_{hyd}$ with $\alpha = 2$ in parameterization P7 resulted in a simulated nocturnal aggregate mean $V_d(NO_2)$ of $0.24 \pm 0.04$ cm s$^{-1}$..."*

**(changed in Section 3.3.3)** *"Physical justification for the scale factor value $\alpha = 2$ necessary to match simulated with observed nocturnal mean $V_d(NO_2)$ could stem from a larger surface area..."* **changed to (L565)** *"Physical justification for a scale factor value $\alpha > 1$ being necessary to reduce the bias between simulated nocturnal $V_d(NO_2)$ and observation-inferred values corrected for estimated soil NO could stem from a larger surface area..."*

**(changed in Section 3.3.3)** *"... in our top-down optimization of $R_c(NO_2)$..."* **changed to (L588)** *"...in our top-down sensitivity analysis of $R_c(NO_2)$..."*

**(changed in Section 3.3.3)** *"We acknowledge that the nocturnal canopy environment to which we optimize simulated $V_d(NO_2)$ is under reduced turbulent mixing..."* **changed to (L590)** *"We acknowledge that the nocturnal canopy environment is under reduced turbulent mixing..."*

**(changed in Conclusions)** *"We recommend the implementation of a mechanistic non-stomatal dry deposition scheme for NO₂ ..."* **changed to (L892)** *"We recommend consideration of a non-stomatal dry deposition scheme for NO₂ ..."*

(2) pointed out that the surface area scale factor $\alpha = 2$ was arbitrarily chosen as part of a sensitivity study, not fit to observations, and is likely a maximum estimate for mature forests:

**(added to Conclusions, L908)** *"We caution that the value $\alpha = 2$ is not a fit to observations at Harvard Forest, rather, a likely upper estimate from a sensitivity study. Long-term field studies quantifying atmosphere–surface exchange across a variety of land types and seasons would facilitate further development of this unique dry deposition pathway".*

(3) added a recommendation for future work in the last sentence of Section 3.3.4:

**(added to Section 3.3.4, L707)** *"Future work investigating the relative roles of R1 and reduction reactions in NO₂ uptake to a variety of natural land type surfaces would aid mechanistic realism in future model developments."*

It is also noted that results from some of the conducted tests are well-known and expected as there are many similar model intercomparison and/or sensitivity studies in literature.

Response: We address this comment in the Response* below.

This is a very long manuscript with too many unnecessary detailed descriptions. While I appreciate the big amount of the work the authors have done, I do feel that it can be substantially simplified because many materials presented here are available in literature and do not necessarily provide any new information.

Response: We appreciate this point of view and have taken steps outlined below to incorporate suggestions for restructuring. Overall, we removed ~5 pages worth of text in this restructuring, which is nearly 20% of the original manuscript content.

For example, the following places can be substantially simplified:

(1) Discussions related to the uncertainties in Ra and Rb (section 3.1 and proceeding sections related to 3.1) have been published in several studies (Toyota et al., 2016, Atmos. Environ., 147, 409–422; Wu et al., 2018, Geoscientific Model Development, 14, 5093–5105). In fact, the same

data set of Nguyen et al. (2015), as used here for testing $R_a$ and $R_b$, was also used in Wu et al. (2018) for the same purpose. Results from these earlier studies are readily applicable here.

Response*: We are aware of Toyota et al. (2016), a very thorough analysis of sensitivity of $R_a$ to model formulation under stable conditions and implications for dry deposition in models. Our manuscript references Toyota et al. (2016) in (i) the Introduction **(L62)** as a reference for the impact of $R_a$ on $V_d$ for very stable conditions and (ii) in what is now Section S1 of the supplement as an authoritative reference for the formulation of $R_a$ in Eq. (S1). However, we are not aware of a study that performs serial quantification of the variables examined sequentially in our parameterizations P1 through P4, and we feel this provides some utility for the modeling community.

To reduce unnecessary detailed descriptions in our discussion of $R_a$, we have removed the following text from Section 3.1:

**(changed in Section 3.1)** "*Referencing $R_a$ from the CTR measurement height of 22 m in parameterization P2 results in a 42 % decrease in $R_a$ under neutral conditions (Fig. S2) and a 23 % decrease under daytime (10–15 LST) conditions (Table 3, P1b vs P2). It should be noted that due to significant contributions of $R_b$ to the total resistance pathway for rapidly depositing species (Table 3, Section 3.2), referencing $R_a$ from GEOS-Chem grid-box-center instead of measurement height, as is commonly done in studies comparing deposition velocities from CTMs to measured values (Clifton et al., 2017; Nguyen et al., 2015; Nowlan et al., 2014; Silva and Heald, 2018), results in a moderate (8 %) decrease in $V_d$ for the species of Table 3 (P5, data not shown).*" **changed to (L331)** "*Referencing $R_a$ from the CTR measurement height of 22 m in parameterization P2 results in a 23 % decrease in daytime $R_a$, returning $V_d$ biases to P1 levels (Table 3).*"

To reduce unnecessary detailed descriptions in our discussion of $R_b$, we have removed the following text from Section 3.2:

**(removed from Section 3.2)** "*As noted by Nguyen et al. (2015), the practice of setting $V_d$ for rapidly depositing species equal to $V_d(HNO_3)$ neglects species-specific diffusion limitations, which can be important under turbulent conditions when $R_a$ is at a minimum. For example, $R_b$ for isoprene nitrate is estimated to be 23 % greater than for $HNO_3$, translating to a -12 % bias in*

$V_d$ under the median midday conditions at the HFEMS ($R_a$(60 m) = 8.6 s m$^{-1}$, $R_b$(HNO$_3$) = 12 s m$^{-1}$; Fig. S6).*"*

We updated the manuscript and referenced Wu et al. 2021, GMD, 14, 5093–5105 in the following:

**(added to Introduction, L79)** *"Using the eddy covariance flux dataset from Nguyen et al. (2015), the Z03 scheme was extended by Wu et al. (2021) to additional species by fitting non-stomatal uptake of oxidized VOCs and hydrogen cyanide directly from observations. Wu et al. (2021) maintain the Z03 algorithm structure through similarity to SO$_2$ and O$_3$; however, they suggest that future developments to dry deposition schemes consider other species-specific processes and reactions affecting measured uptake, including below-sensor chemical flux divergence, enzymatic reactions, and other non-stomatal processes/reactions."*

**(moved from Section 2.2.1 to Section 2.2, L163)** *"...being able to neglect the complexities of a surface resistance scheme allows for a more direct evaluation of $R_a$ and $R_b$ components of the resistance-in-series pathway used in the parameterization of $V_d$ (Wu et al., 2021)."* We added Wu et al. (2021) as a reference for this strategy.

**(added to Section 3.1, L316)** *"...by comparing to measured daytime deposition velocities for rapidly depositing species from Nguyen et al. (2015). In contrast to the findings of Wu et al. (2021) showing excellent model-measurement agreement to peak daytime $V_d$(HNO$_3$) and $V_d$(H$_2$O$_2$) between the Z03 scheme and the dataset from Nguyen et al. (2015), parameterization P1 overestimates ..."*

We would like to add that Zhiyong Wu's work, namely, Wu et al., 2011, Atmos. Environ., 45(16), 2663-2674 which evaluated simulated $V_d$ for O$_3$ and NO$_y$ from stand-alone dry deposition modules from two community models against the Harvard Forest dataset, was inspirational in the planning of this manuscript. In addition to the existing references to Wu et al. (2011) in Section 3.4, we have added the following:

**(added to Section 2.2.2, L204)** *"..., following the approach of Wu et al. (2011)..."*

**(added to Section 2.2.2, L223)** *"..., including $z_o$ for Harvard Forest (Wu et al., 2011)."*

The work of Wu et al., 2018, J. Adv. Model Earth Syst., 10(7), 1571-1586, which evaluates and intercompares simulated $V_d$ for $O_3$ and $SO_2$ from five dry deposition algorithms against observations from Borden Forest, is now referenced in the revised manuscript as it relates to our implementation of the Zhang 2003 (Z03) dry deposition algorithm:

**(Section 2.1.2, L155)** *"Following a similar approach to Wu et al. (2018), we compute component surface resistances for the mixed forest as an average of deciduous broadleaf and evergreen needleleaf land type specific values from Z03, weighted by LAI-determined deciduous and coniferous fractions for Harvard Forest..."*

(2) Details of all the measurement data and model formulas (Section 2) are available from literature; they can either be referred to the literature and/or moved to Supplement Information of this manuscript. Only keep the minimum information that is needed for the big picture.

Response: Agreed. The revised manuscript implements these recommendations in Section 2. Changes made in Section 2 relating to model descriptions:

**(moved to supplement)** Section *"2.1 Trace gas dry deposition parameterization from the GEOS-Chem model"* was moved to Section S1 of the supplement. Equation numbers were updated accordingly.

**(title change, Section 2.1, L133)** Section 2.1 in now titled *"2.1 Reference algorithms for computing gaseous dry deposition velocities"*

**(sub-section added, Section 2.1.1, L134)** Sub-section *"2.1.1 GEOS-Chem dry deposition module"* is a short paragraph containing high-level descriptions and pointers to supplemental material and literature.

For eddy covariance measurement descriptions in Section 2, we removed details redundant to references provided, but keep high-level site descriptions and details/methods important for understanding (i) how we filter the measurement data and (ii) representation in simulations. Specific changes made to Section 2 relating to measurement descriptions:

**(removed from Section 2.2.1)** (i) *"... with large forest fetch to the N, W, and E ..."*

(ii) *"... from a walk-up tower, with sonic anemometer (8 Hz) and inlet to the time-of-flight (TOF) chemical ionization mass spectrometer (CIMS) (10 Hz) facing north so as to capture eddies originating over forest fetch."*

(iii) *"...($6^{th}$, $15^{th}$, $20^{th}$, $23^{rd}$, and $27^{th}$)..."*

**(removed from Section 2.2.2)** (i) *"Local pollution sources include a secondary paved road 1.5 km to the west, a two-lane expressway ~ 5 km to the north, and a small town more than 10 km to the northwest."*

(ii) *"Briefly, $NO_y$ concentrations at 8 Hz were made by reducing $NO_y$ to NO on a well-aged hot gold catalyst with $H_2$, followed by detection of chemiluminescence from titration of resulting NO with $O_3$. The reducing catalyst was positioned close to the inlet at a height of 29 m on the 30 m walk-up tower. Concentration measurements of PAN by capillary-column gas chromatograph with electron capture detection was added to the 30 m walk-up tower in April 2000 (Horii et al., 2005). To an auxiliary 23 m scaffolding tower located ~ 100 m to the southeast of the main tower, a Tunable Diode Laser Absorption Spectrometer (TDLAS) was configured to measure eddy covariance fluxes (1 Hz) of $NO_2$ and concentrations of $HNO_3$ from April through November 2000 (Horii et al., 2004). Due to inlet wall interactions of $HNO_3$ with a characteristic time constant of ~ 10 minutes, high frequency concentration information required for eddy covariance flux computation was not possible; however, it was found that the hourly mean concentration was not compromised, as the fluorinated silane-coated fused silica quartz inlet walls were not a permanent sink of $HNO_3$ which was near completely transmitted to the measurement cell after sufficient equilibration time (Horii et al., 2005). Although the measurement height of $NO_y$, NO, $NO_2$, and PAN on the main tower (29 m) did not match that of $HNO_3$ and $NO_2$ on the auxiliary tower (22 m), Horii et al. (2005) found the measurement heights of the two towers to be in the same flux regime by congruence of heat fluxes and noted as well the coherence in coincident trace gas data on the hourly timescale."* **was replaced with (L193)** *"Measurements of above-canopy PAN concentrations were added in April 2000 (Horii et al., 2005). Eddy covariance fluxes of $NO_2$ along with above-canopy (22 m) measurements of $HNO_3$ concentrations were made at the HFEMS from April through November 2000 (Horii et al., 2004)."*

(iii) *"Excellent agreement (normalized mean bias NMB < 1 % and $R^2$ > 0.93) is noted for T and P; RH and $u_*$ have small biases of -6 % and 5 %, respectively, with an $R^2$ of 0.60 and 0.69,*

*respectively. Sensible heat flux and downward shortwave radiation are each biased high by 16 %, with an $R^2$ of 0.68 and 0.87, respectively."* **was replaced with (L220)** *"..., depicting good to excellent agreement."*

**(removed from Section 3.3.1)** *"Although we report biases between simulated and observation-inferred $V_d(NO_2)$ using outlier-filtered mean values, median and $\overline{F}/\overline{[NO_2]}$ values of $V_d(NO_2)$ are included in Table 4 for comparison."* This was repeating statements already made and obvious by consulting Table 4.

**(changed in Section 3.3.1)** *"Herein, we restrict analysis to average values over at least one month, as simple resistance-in-series parameterizations of dry deposition employing 'big leaf' representations of $R_c$ are designed for computational expediency, general applicability over a wide range of land types, and to reflect average estimates over weeks to months and therefore lack the necessary complexity to capture the full range of short-term variability at specific sites (Wesely, 1989)."* **changed to (L489)** *"Herein, we restrict analysis to average values over at least one month."*

(3) Do you really need formulas and detailed descriptions of Vd models in Section 1? Should you focus more on NO2 hydrolysis effect on Vd, instead of describing Vd modes similar to a textbook?

Response: Although equations (1) & (2) in the Introduction are well-known, we would prefer to keep them, along with their high-level descriptions, since they are foundational to both observation-inferred and simulated $V_d$. We also would prefer to keep the paragraph describing the Wesely 1989 and Zhang 2003 parameterizations of $R_c$ for use in CTMs since a main objective of the manuscript is to explore a deviation from these algorithms for the purpose of representing $NO_2$ hydrolysis through non-stomatal uptake. Focusing more on $NO_2$ hydrolysis effect on above-canopy $V_d(NO_2)$ would be difficult in the Introduction since, to our knowledge, this is a first attempt to quantify this effect in a dry deposition scheme, although it has certainly been speculated about in previous works, as referenced in the Introduction. Instead, we discuss the breadth of laboratory and field observations that provides the foundation enabling this work. We have, however, removed the following sentence from the Introduction that we feel distracts from the manuscript's objective:

**(removed from Introduction)** *"Flechard et al. (2011) compare dry deposition fluxes of reactive nitrogen species NH₃, NO₂, HNO₃, HONO, particulate ammonium (pNH₄) and nitrate (pNO₃) across an inferential network of 55 sites throughout Europe using four existing dry deposition routines and note differences between models (up to a factor of 2 to 3) are often greater than differences between sites, calling for more long-term direct $N_r$ flux measurements with which to validate dry deposition algorithms."*

(4) You can simply state that you used a stand-alone version of GEOS-Chem Vd code in this study for sensitivity tests, why bother describing how you extract the code from GEOS-Chem CTM (in Abstract, Introduction, section 2.1)?

Response: Agreed. We have made the following changes:

**(changed in Abstract)** *"... by extracting the trace gas dry deposition algorithm and reimplementing in ..."* **changed to (L11)** *"... by running a stand-alone version of $V_d$ code ..."*

**(changed in Introduction)** *"In this study, we extract the trace gas dry deposition parameterization from the GEOS-Chem global CTM and reimplement to run in single-point-mode to facilitate evaluation of an updated parameterization that includes the effect of reaction R1 on simulated $V_d(NO_2)$ and $V_d(NO_y)$. We compare to above-canopy observations ..."* **changed to (L122)** *"In this study, we compare simulated dry deposition velocities from the GEOS-Chem CTM to above-canopy observations ..."*

**(changed in Section 2.1)** *"... we extract the trace gas dry deposition source code and input parameters from GEOS-Chem v10-01 (www.geos-chem.org) and implement in single-point-mode ..."* **changed to (L135)** *"...a stand-alone version of the gaseous dry deposition algorithm from GEOS-Chem v10-01 (www.geos-chem.org), implemented to run in single-point-mode..."*

**(changed in Conclusions)** *"Extraction of the trace gas dry deposition algorithm from GEOS-Chem and reimplementation to run offline in single-point-mode enabled..."* **changed to (L861)** *"A stand-alone version of the gaseous dry deposition algorithm from GEOS-Chem, implemented to run in single-point mode, enabled..."*

The modification of GEOS-Chem Vd code by including surface NO2 hydrolysis pathway increases NO2 Vd (probably because the original Wesely code predicts very low Vd for NO2).

Response: Yes, but we note a main objective of the manuscript was to investigate to what extent $NO_2$ hydrolysis could explain observation-inferred nocturnal $V_d(NO_2)$, assuming negligible contributions from other reactions.

If it is not too much work, can the authors estimate how higher the increased NO2 Vd would be compared to other Vd models such as the Zhang et al. (2003) code that is referenced here?

Response: We think this is a very good point and well-worth the effort to include in the revised manuscript. To compare to nocturnal observations of $V_d(NO_2)$, we implemented the non-stomatal branch of the Zhang 2003 dry deposition algorithm (Z03) in parameterization P5 in the revised manuscript. Updates to Tables and Figures include:

**(Table 2, ~L310)** Parameterization P5 added to the list of serial modifications made.

**(Figure 3, ~L470)** Simulated nocturnal $V_d(NO_2)$ following Z03 scheme, P5(Z03), depicted alongside parameterizations P4, P6, & P7.

**(Table 4, ~L491)** Parameterization P5 included in list of simulated campaign mean and median nocturnal $V_d(NO_2)$.

**(Figure 4, ~L675)** Simulated nocturnal $V_d(NO_2)$ following Z03 (parameterization P5) is now included in panels depicting monthly mean parameterized nocturnal $V_d(NO_2)$ from bottom-up and top-down approaches.

**(Table S3 of supplement)** Monthly mean nocturnal canopy surface resistances $R_c$ and resulting $V_d(NO_2)$ computed following Z03 scheme added under heading *Z03 $V_d(NO_2)$*.

Text added to revised manuscript to describe implementation of Z03 and discussion of results:

**(sub-section added, Section 2.1.2, L145-159)** Sub-section "*2.1.2 Non-stomatal branch of Z03 dry deposition algorithm*" recaps a brief introduction to the Z03 scheme with specifics related to non-stomatal treatment of $NO_2$. It then describes our methodology of implementation, specifically how we treat canopy wetness, snow cover fraction, and mixed forest type.

**(changed in Section 3)** "*...,updating non-stomatal surface resistance for $NO_2$ to include heterogeneous hydrolysis on deposition surfaces (P6, P7), ...*" **changed to (L274)** "*..., updating non-stomatal surface resistance for $NO_2$ following the Z03 scheme (P5) and subsequent*

*replacement with a scheme that represents heterogeneous hydrolysis on deposition surfaces (P6, P7), …"*

**(changed in Section 3.3.1)** *"We begin discussion of eddy covariance observed bulk-canopy $V_d(NO_2)$ below, followed by discussion in Section 3.3.2 of the bias in simulated values stemming from the widely used W89 parameterization of surface resistances."* **changed to (L461)** *"We begin discussion of eddy covariance inferred bulk-canopy $V_d(NO_2)$ below, followed by discussions in Section 3.3.2 of the (i) large low bias in simulated values stemming from the widely used W89 parameterization of surface resistances and (ii) reduced bias using the Z03 scheme."*

**(sub-section title change, Section 3.3.2)** *"3.3.2 Evaluation of parameterized $V_d(NO_2)$ from GEOS-Chem"* **changed to (L506)** *"3.3.2 Evaluation of nocturnal $V_d(NO_2)$ from GEOS-Chem and Z03"*

**(added to Section 3.3.2, L512)** *"Wesely et al. (1982) reported a nocturnal eddy covariance observed $V_d(NO_2)$ of 0.05 cm $s^{-1}$ over a summertime soybean field, similar to the P4 value in Table 4. The authors acknowledge that counteracting soil NO emissions may have resulted in low measured values of above-canopy $NO_2$ deposition."*

**(paragraph added to Section 3.3.2, L520-532)** This paragraph discusses $V_d(NO_2)$ simulated following the Z03 scheme, as it appears in Fig. 3 and Table 4 (i.e., comparison to eddy covariance inferred values). The paragraph notes good model-measurement agreement, much better than with the W89 scheme. The paragraph also discusses the reasons behind W89 and Z03 differences.

**(paragraph added to Section 3.3.2, L533-539)** This paragraph discusses the reason that the Z03 scheme, although in good agreement with observations, could be a potential misrepresentation of non-stomatal $NO_2$ dry deposition, i.e., it is based on suspected $NO_2$ reduction reactions relative to $O_3$ dry deposition and does not consider $NO_2$ surface hydrolysis. The paragraph notes that the Z03 scheme was optimized to field observations for $NO_2$ dry deposition, studies which were also referenced in our manuscript when exploring the potential role of $NO_2$ hydrolysis on $V_d(NO_2)$.

**(paragraph added to Section 3.3.4, L699-708)** This paragraph discusses $V_d(NO_2)$ simulated following the Z03 scheme, as it appears in Fig. 4 (i.e., a twelve-month comparison to bottom-up $V_d(NO_2)$ computed using $NO_2$ uptake coefficients from Table 1). It is noted that simulated monthly values following Z03 are in good agreement with both top-down estimates and bottom-up estimates for the reduced bark uptake case (i.e., Fig. 4, bottom panel). It again is noted that Z03 sets non-stomatal $NO_2$ uptake with reference to field observations, and that our manuscript sought to explore the plausibility that $NO_2$ hydrolysis could explain non-stomatal uptake (assuming negligible contributions from other reactions) to the natural land types examined and reviewed in Sections 3.3.3 & 3.3.4. We acknowledge that other reactions are expected to make important contributions to $NO_2$ uptake in some instances, and we point to Section S6 of the supplement which discusses $NO_2$ uptake within leaf interiors rich in antioxidants. We conclude this paragraph with an acknowledgement that *"Future work investigating the relative roles of R1 and reduction reactions in $NO_2$ uptake to a variety of natural land type surfaces would aid mechanistic realism in future model developments."*

**(added to Conclusions, L890)** *"Although we find that the Z03 dry deposition scheme adequately captures the magnitude of nocturnal $V_d(NO_2)$ over Harvard Forest, formulating uptake relative to $V_d(O_3)$ neglects contributions from reaction R1, and therefore may be a misrepresentation for $NO_2$ with implications for surface HONO production in models.."*

In addition to the above descriptions of implementation and discussion of Z03, we utilized parameterizations from the Z03 scheme in Section 3.3.4 for (i) dew wetted canopy and (ii) snow cover fraction so as to have consistent representations when comparing to parameterized top-down and bottom-up $V_d(NO_2)$ in Fig. 4 and Table S3. These modifications result in small changes to bottom-up and top-down $R_c(NO_2)$ & $V_d(NO_2)$ for winter months, and small changes to bottom-up $R_c(NO_2)$ & $V_d(NO_2)$ for spring, summer, and fall months (Fig. 4 & Table S3 updated accordingly), but not enough to change our discussion of results or conclusions. The following modifications were made to method descriptions in the revised manuscript:

**(changed in Section 3.3.4)** *"...which lack stomatal pores and therefore the elevated water vapour concentrations sufficient to support thin water films for reaction R1 to proceed under low to moderate ambient RH. The forest canopy is expected to be dew covered under conditions of high ambient RH > 96 % (Turnipseed et al., 2006), at which point we assume both top and*

*bottom faces of deciduous leaves would support thin water films; we therefore increase the α value used to scale uptake to deciduous leaves from LAI to 'total leaf area' (i.e., 2LAI) for RH > 96 %. As seen in Table 1, uptake to wet bark is twice that of dry bark, and we assume wetted bark for RH > 96 %. Also from Table 1, NO₂ uptake to snow is approximately one-third that of the forest floor; we make the assumption that the forest floor is snow covered in winter months (DJFM) for ambient temperatures < 0 °C."* **changed to (L655)** *"...which lack the elevated water vapor concentrations from stomata to support thin water films for reaction R1 to proceed. We treat the leaves and bark of the forest canopy as wet following the dew flag from the Z03 scheme (Section 2.1.2). Under wet conditions, both top and bottom faces of deciduous leaves would be wetted and we increase the α value used to scale uptake to deciduous leaves from LAI to 2LAI. The α value used to scale uptake to bark is computed as πSTAI, where STAI = 0.9 is the projected area of tree branches (Horii et al., 2005). We compute forest floor surface resistances $r_c$ floor as parallel contributions from uptake to snow and the snow-free forest floor, weighted by snow cover fraction following the Z03 scheme (Zhang et al., 2003a). As seen in Table 1, NO₂ uptake to wet bark is twice that of dry bark, and uptake to snow is approximately one-third that of the snow-free forest floor.*

We removed a paragraph from Section 3.3.3 that became redundant to the above added descriptions and discussions of the Z03 scheme and its comparison to the other parameterizations in this study:

**(removed from Section 3.3.3)** *"By replacing the non-stomatal pathways of NO₂ deposition from the W89 algorithm with $r_{hyd}$ according to Eq. (11), we assume that non-stomatal deposition of NO₂ is due entirely to heterogeneous hydrolysis following reaction R1. Zhang et al. (2003) neglect solubility contributions to NO₂ uptake in their dry deposition scheme, relying entirely on similarity to O₃ reactivity. The W89 dry deposition scheme assigns NO₂ to the 'slightly reactive' category, intended for substances with limited biological reactivity but still requiring very small leaf mesophyll resistances so that NO₂ deposits under stomatal control. This classification for NO₂ in the W89 scheme results in near-negligible non-stomatal deposition, yielding a non-stomatal $R_c(NO_2)$ of ~ 2,700 s m⁻¹ at Harvard Forest—much above observation-inferred values, as previously discussed. Adding $r_{hyd}$ in parallel to the W89 non-stomatal deposition pathway, instead of in replacement of, results in a slight increase (~ 10 %) in simulated mean $V_d(NO_2)$*

*over Harvard Forest for parameterization P7—still supporting P7 with α > 1, but possibly not as large as α = 2. Variability in observed $V_d(NO_2)$ and uncertainties in the assumption of a non-stomatal $R_c(NO_2)$ pathway following similarity to $SO_2$ and $O_3$ uptake make more precise recommendations difficult."*

How big differences would such a modification of NO2 Vd make on predicted ambient concentration and deposition flux of NO2 on seasonal to annual basis?

Response: The effect of updates on inferred campaign-mean deposition flux of $NO_2$ over Harvard Forest are depicted in Table 5 and discussed in Section 3.4.1, i.e., 56% increase in inferred 24 h $NO_2$ deposition flux due to $NO_2$ hydrolysis updates **(L770)**. Although this manuscript isn't able to comment on changes to simulated $NO_2$ concentration fields, we **updated (~L491) Table 4 to include NO₂ lifetime to dry deposition** to give a more intuitive sense of how $V_d(NO_2)$ updates would affect concentrations. Accordingly, Table 4's introduction was updated in the revised text:

**(changed in Section 3.3.1)** *"Table 4 presents observed and simulated values of $V_d(NO_2)$ aggregated across all months, as well as associated canopy reduction factors used to correct observed $V_d(NO_2)$ for soil NO."* **changed to (L459)** *"Table 4 presents observation-inferred and simulated values of $V_d(NO_2)$ aggregated across all months, as well as associated $NO_2$ lifetimes to dry deposition from the 29 m measurement height."*

We have also updated the concluding sentence of the manuscript to provide more information on what will be included in forthcoming work:

**(changed in Conclusions)** *"... including analysis of the resulting HONO simulation."* **changed to (L905)** *"...including analysis of impacts on simulated concentration fields (i.e., $NO_2$, HONO, OH, and $O_3$) and deposition budgets."*

Overall, this is a big effort and can be published after considering the above comments.

Response: We thank Reviewer 1 for their helpful feedback which we believe has improved both the utility and focus of this manuscript.

**RC2**

**Summary**: In this manuscript the authors test the sensitivity of GEOS-Chem assumptions about gas-phase diffusivity, site-specific parameters that influence the $R_a$ parameter in the Wesley gaseous dry deposition model, and non-stomatal influences on $R_c$. They use a dataset collected in 2000 from Harvard Forest to evaluate their non-stomatal treatment of $R_c$ for $NO_2$ deposition. They then consider a set of various parameterizations that implement different treatments of parameters that affect $R_a$, $R_b$, and $R_c$ for $NO_2$ in GEOS-Chem. Informed by the results of these updated parameterizations, the authors improve agreement between modeled and measured $V_{dep}$ for highly soluble species measured from the study of Nguyen et al. 2015. Among other things, they also highlight the likely importance of heterogenous hydrolysis of $NO_2$ as a deposition mechanism.

I agree with the perspective provided by Reviewer 1 that the manuscript could benefit from condensing some of the sections that have a lot of detail and/or read like a textbook. I found the manuscript difficult to review because the amount of detail that was included about the analyses.

Response: In addition to length edits following RC1 suggestions, we have removed from the revised manuscript parameterization P4, which wasn't a logical progression in the thought process to address bias in rapidly depositing species. Rather, it was simply there to point out a thought experiment tangential to Section 3.1. Although it only removes a single sentence from Section 3.1, we feel it improves readability by not disrupting the objective behind serial updates in Table 4.

**(removed from Section 3.1)** *"Parameterization P4 in Table 3 shows the effect of incorrectly neglecting the non-zero wind at $z_o$ in the RSL correction of $R_a$, resulting in a 50 % reduction in $R_a$ and a significant increase to the high biases in $V_d$ for the rapidly depositing species of Table 3."*

Accordingly, we have removed depictions of old parameterization P4 from associated tables (Tables 2 & 3) and figures (supplemental Figs. S1), and updated parameterization numbers in the revised manuscript throughout.

To further reduce length and unnecessary details that detract from the primary focus of the manuscript, we have **moved** Section *"3.4.1 Simulated diel profiles of $V_d(x)$ for measured $NO_y$ component species"* and associated Fig. 6 **to Section 5 of the supplement**. Section and figure

numbers were updated accordingly. This section mostly reviewed parameterization updates already discussed in previous sections as they applied to simulated diel $V_d(x)$ used to calculate simulated $V_d(NO_y)$ in Eq. (11). Original Fig. 7 (now Fig. 6) applies the main contents of original Fig. 6 (now Fig. S4), which are already discussed in what is now Section 3.4.1 **(L739)** *"3.4.1 Measurement–model comparison of $V_d(NO_y)$"*. To direct interested readers while preserving the main focus of the manuscript, we add the following succinct summary:

**(added to Section 3.4, L735)** *"Diel profiles of simulated $V_d(x)$ used in Eq. (11) to compute simulated $V_d(NO_y)$ are depicted in Fig. S4 and reviewed in Section S5 of the supplement. Given the predominant contributions that $NO_2$ and $HNO_3$ make to the $NO_y$ budget at Harvard Forest (Fig. 5), updates to the parameterization of $V_d(NO_2)$ and $V_d(HNO_3)$ discussed in previous sections will have a notable impact on simulated $V_d(NO_y)$."*

To further aid readability, we have removed/changed the following sentences that contain unnecessary detail and/or are tangential to the manuscript's main objectives:

**(removed from Section 3.1)** *"Given the lower relative measurement height at the HFEMS (~ 1.5 $h_c$), P3 $R_a$ results in increases over P2 $R_a$ by 30 % (10[th] percentile), 20 % (50[th] percentile) and 18 % (90[th] percentile) at this site (Fig. S1); simulated $V_d(HNO_3)$ over Harvard Forest is discussed in Section 3.4.1."*

**(changed in Section 3.2)** *"...consistent with the findings of Tang et al. (2014 & 2015) from an evaluation of a comprehensive diffusivity dataset of atmospherically relevant reactive trace gases for which Tang et al. have made the results publicly available."* **changed to (L378)** *"...consistent with the findings of Tang et al. (2014 & 2015)."*

**(removed from Section 3.2)** *"Referencing Graham's law from measured $D_{H_2O_2}$, as done in Nguyen et al. (2015), degrades comparison to inorganic diffusivities (NMB ~ -14 %), improves comparison to organics (NMB < 2 %), and has no effect on correlation ($R^2$ = 0.91) (data not shown). Sensitivity of Graham's law to choice of reference species is not surprising given the deviation of the $\sqrt{M_2/M_1}$ dependence from the functional form of Eq. (8)."*

**(changed in Section 3.3.2)** "...*increasing to a 7-fold low bias after correcting for soil NO with a corresponding simulated CRF from parameterization P5 of 31% (Table 4)."* **changed to (L509)** "...*increasing to a 7-fold low bias after correcting for soil NO from parameterization P4 (Table 4)."*

**(removed from Section 3.3.3)** *"Although chamber studies generally observe first-order uptake of $NO_2$ under controlled conditions, constant $V_d(NO_2)$ inferred from linear regression of eddy covariance fluxes of $NO_2$ versus concentration is not expected due to variability in turbulence and surface conditions affecting uptake (i.e., surface wetness, stomatal aperture, and surface area)."* This sentence was an unnecessary detail given the explanatory sentence that follows it: **(L606)** *"By restricting analysis to nocturnal conditions when..."*

**(changed in Section 3.3.3)** *"Plake et al. (2015) find a maximum median nocturnal bulk $R_c(NO_2)$ over a natural grassland site in Mainz, Germany (August–September) of 560 s $m^{-1}$ via the dynamic chamber approach, attributing all flux of $NO_2$ to deposition since soil NO emissions for this nutrient poor site were below chamber detection limits."* **changed to (L628)** *"Plake et al. (2015) find a maximum median nocturnal bulk $R_c(NO_2)$ over a natural grassland site in Mainz, Germany of 560 s $m^{-1}$ via the dynamic chamber approach."*

**(removed from Section 3.3.4)** *"Meteorological and phenological data required to compute thermal speed $\overline{v}_t$ and surface area scale factor α in Eq. (11) are from observations (i.e., Fig. S4 & S5)."* and *"For complex surfaces with extensive and difficult to quantify substructures, planar (i.e., forest floor) or geometric (i.e., tree bark) surface areas are used. For more tractable surfaces such as deciduous or coniferous foliage, corresponding total leaf area or LAI are used (i.e., Fig. S4)."* These sentences were redundant to the information presented in Table 1, Table S3, Section 2.2.2, and the sentence preceding them **(L647):** *"Component surface area scale factors α are material dependant, varying according to the surface area used to normalize deposition fluxes in corresponding measurement studies (Tables 1 & S2)"*

**(removed from Section 3.3.4)** (i)*"...a 27 % difference over the 12-month period when top-down $V_d(NO_2)$ is computed using α = 2, increasing to an 87 % difference for α = 1."*

(ii) *"...contributing about half the total nocturnal uptake across all seasons with the remaining half apportioned between the forest floor (29 %) and foliage (21 %)."*

(iii) *"...(shagbark hickory, tulip poplar, loblolly pine, and southern red oak)..."*

These statements provided extra details to discussions of the middle panel of Fig. 4, which we argue in Section 3.3.4 could overestimate $NO_2$ uptake to bark given a likely mismatch between average bark surface area at Harvard Forest and chamber measurements from literature: **(L688)** *"Bark surface area for the trunk and branch samples examined in Hanson et al. (1989) may not be representative of the average bark surface area for the canopy at Harvard Forest."* As such, we removed these unnecessarily detailed statements tangential to the main point that there's good reason uptake to bark should be reduced by half.

**(changed in Section 3.4)** *"Of particular interest is the period from June–November 2000, when hourly observations of above-canopy $HNO_3$ concentration—a significant contributor to $NO_y$ dry deposition at this location (Horii et al., 2005)—was added to the suit of long-term measurements which include hourly concentrations of total $NO_y$ and component species NO, $NO_2$, and PAN alongside eddy covariance measurements of $NO_y$ flux (Fig. S5). The top panel of Fig. 5 depicts the diel climatology (June–November 2000–2002) of measured $NO_y$, NO, $NO_2$, $HNO_3$, and PAN over Harvard Forest. Also depicted is inferred $NO_y$ computed as the sum of aforementioned component species."* **changed to (L713)** *"Of particular interest is the period from June–November 2000, when hourly observations of above-canopy $HNO_3$ concentration—a significant contributor to $NO_y$ dry deposition at this location (Horii et al., 2005)—was added to the suit of measurements (Fig. S5). The top panel of Fig. 5 depicts the diel climatology of measured $NO_y$, and constituent species over Harvard Forest, alongside inferred $NO_y$ computed as the sum of the component species."*

**(changed in Section 3.4)** *"...where the diel climatologies of component fractions $\overline{[x_i]}_h^{meas} / \overline{[NO_y]}_h^{meas}$, depicted in the middle panel of Fig. 5, are computed over June–November 2000–2002 and subjected to a smoothing spline fit."* **changed to (L729)** *"...where the diel climatologies of component fractions $\overline{[x_i]}_h^{meas} / \overline{[NO_y]}_h^{meas}$ are depicted in the middle panel of Fig. 5."* Removed information is a repetition (Fig. 5 caption and later beginning on **L731** *"Here, we compute component fractions as the 'ratio of smoothed means'..."*

**(removed from Section 3.4)** *"A large fraction of inferred values for PAN results from the absence of observations from August–November 2000, thus relying on years 2001–2002 to*

*inform the climatology using Eq. (12) (Fig. S6). HNO₃ was also inferred to a large degree; although hourly concentrations were measured fairly consistently from June–November 2000 (Fig. S6), monthly coverage was only ~ 20 % (Fig. 5).*" This detailed information may be disruptive to narrative flow and not overly useful for interpretation at the location it was presented. In Section 3.4.1, a comment on $HNO_3$ measurement coverage is made in relation to discussing a specific model-measurement disagreement in Fig. 6., i.e., **L763** *"Given the heavy reliance on the climatological diel profile of HNO₃ (Fig. 5) due to the paucity of hourly HNO₃ observations (Fig. 5 & Fig. S6)..."*

**(removed from Section 3.4.2)** (i) *"By monitoring the rate of change of vertically integrated in-canopy concentration profiles, Munger et al. (1996) showed that canopy storage contributions to above-canopy NO$_x$ and O$_3$ fluxes were small at Harvard Forest; however, as noted by Horii et al. (2005), NO$_y$ fluxes measured at the HFEMS did not include a storage term as canopy vertical profiles of NO$_y$ were not available."*

(ii) *"Future analysis of this measurement–model discrepancy would benefit from in-canopy vertical profiles of NO$_y$, particulate speciation, and component species of both at hourly resolution."*

Sentences (i) and (ii) are tangential details that stray beyond the manuscript's scope of analysis *"Given the heavy reliance on the climatological diel profile of HNO₃ (Fig. 5) due to the paucity of hourly HNO₃ observations (Fig. 5 & Fig. S6), and the potential influence of unmeasured rapidly depositing NO$_y$ species (Horii et al., 2005),..."* **(L763)**

However, this was a monumental effort, and the authors methods of analyses are scientifically sound and robust. I left some comments below that should not be necessary to address but could be considered in constructing future manuscripts.

**Comments**

**(General)** About halfway through the manuscript I had to remind myself, by re-reading the introduction, what motivated this study. For me, the question of why $NO_2$ and what are CTMs getting wrong about $NO_2$ deposition wasn't really addressed until page 4. Echoing Reviewer 1 I felt like I had a textbook introduction to dry deposition of gases before the uncertainties of $NO_2$ deposition (likely non-stomatal deposition) were introduced. I don't have specific

recommendations for restructuring/condensing the introduction, but it could benefit the reader to be concise in stating the problem that updating the gaseous diffusivity and non-stomatal deposition treatment for NO2 in GEOS-Chem could improve measurement-model agreements.

Response: We agree that concisely stating the manuscripts main findings upfront would help the reader maintain focus. We have incorporated your recommendation in the begging portion of the Abstract:

**(added to Abstract, L12)** *"Improved measurement-model agreements result mainly from (i) updates to the calculation of molecular diffusivities and (ii) representing ground surface $NO_2$ hydrolysis in formulation of non-stomatal uptake."*

We also modify the last sentence of the Abstract to highlight our objective of developing a representation of non-stomatal $NO_2$ uptake that considers $NO_2$ hydrolysis as a surface reaction pathway:

**(added to Abstract, L21)** *"These developments are a first step towards a tractable representation of $NO_2$ hydrolysis in a dry deposition scheme and have important implications for near-surface $NO_2$ lifetime through a mechanism involving HONO emission."*

We also added a synthesis statement at the beginning of Section 3.4 to highlight how the manuscript's main findings are evaluated at a broader scale through an $NO_y$ budget analysis.

**(added to Section 3.4, L710)** *"Improved representation of $V_d$ for $NO_y$ component species $HNO_3$ and $NO_2$ via updates to molecular diffusivity (Section 3.2) and non-stomatal $NO_2$ uptake (Section 3.3.3), respectively, may be further evaluated at a broader scale through a full $NO_y$ budget analysis."*

**(Table 2)** There's a missing ")" for the "base GC (Chapman–Enskog theory with constant mfp(b)" in the middle box of the table.

Corrected.

**(Page 19 Line ~500)** Detailing the historical observations and exact calculation methodology of $V_{chem}$ and $F_{soil}$ is an example of where some information could potentially be condensed.

Response: We would prefer to keep detailed descriptions of $V_{chem}$ and $F_{soil}$ within the revised text, as we think they're useful to understanding our methods. However, we agree with the suggestion that detailed descriptions of historical observations in this section could be condensed. The revised text makes the following changes:

**(changed in Section 3.3.1)** *"Studies have noted the importance of knowledge of local soil NO emissions and within-canopy processes involving $NO_x$ when interpreting above-canopy $NO_2$ fluxes (Delaria and Cohen, 2020; Eugster and Hesterberg, 1996; Flechard et al., 2011; Min et al., 2014). Using measured soil NO emissions from a Ponderosa Pine plantation 75 km from Sacramento, California, Min et al. (2014) calculate an $NO_2$ flux resulting from the reaction of soil NO with $O_3$ to be 3.5 times greater than the observed above-canopy eddy covariance $NO_2$ flux, indicating in-canopy $NO_2$ loss processes which authors mostly attribute to daytime organic nitrate production. In their analysis of eddy covariance fluxes of $NO_2$ over a managed grassland in central Switzerland, Eugster & Hesterberg (1996) found that accounting for counteracting fluxes of soil-emitted NO, oxidized to $NO_2$ below the height of the sensor (~ 2.7 to 3.6 ng N $m^{-2}$ $s^{-1}$), resulted in an increase in inferred nocturnal $V_d(NO_2)$ by up to a factor of 2; sensitivity tests showed a 50 % change in estimated soil NO emission resulted in a change in inferred $V_d(NO_2)$ on the order of 25 %."* **changed to (L422)** *"Previous studies have noted the importance of knowledge of local soil NO emissions and within-canopy processes involving $NO_x$ when interpreting above-canopy $NO_2$ fluxes (Delaria and Cohen, 2020; Eugster and Hesterberg, 1996; Flechard et al., 2011; Min et al., 2014)."*

A small portion of the removed description above was moved to Section 3.3.2 as part of discussion of the W89 low bias. We made the following change in the revised text:

**(changed in Section 3.3.2)** *" In particular, Eugster & Hesterberg (1996) inferred a median value for nocturnal non-stomatal $R_c(NO_2)$ of 700 s $m^{-1}$ (range 500–950 s $m^{-1}$) over a managed grassland in central Switzerland—a surface resistance on the order of 4 times lower than predicted by the W89 algorithm."* **changed to (L514)** *"In their analysis of eddy covariance fluxes of $NO_2$ over a managed grassland in central Switzerland, Eugster & Hesterberg (1996) found that accounting for counteracting fluxes of soil-emitted NO, oxidized to $NO_2$ below the height of the sensor (~ 2.7 to 3.6 ng N $m^{-2}$ $s^{-1}$), resulted in an increase in inferred nocturnal $V_d(NO_2)$ by up to a factor of 2, corresponding to an inferred median value for nocturnal non-*

*stomatal $R_c(NO_2)$ of 700 s $m^{-1}$ (range 500–950 s $m^{-1}$)—a surface resistance on the order of 4 times lower than predicted by the W89 algorithm."*

We have also removed a few sentences from Section 3.4 that were tangential to the description of methods, and redundant to the discussion of results already present as a concluding sentence in Section 3.4, i.e., (**L805**) *"Horii et al. (2005) provide evidence of a rapidly depositing unidentified $NO_y$ species at this site, especially under southwesterly flow, and speculate the unidentified $NO_y$ species as organic nitrates."*.

(**removed from Section 3.4**) *"As discussed in Horii et al. (2005), Harvard Forest is influenced by two predominant airmasses: (i) northwesterly flow brining cool, dry, and less polluted air with an $NO_y$ concentration budget that is mostly closed by $NO_x$, $HNO_3$, and PAN and (ii) southwesterly flow consisting of warmer, humid, and significantly more polluted air wherein up to 50 % of the $NO_y$ budget remains unaccounted for, although the rank of measured contributions remains in the order $NO_2$ > $HNO_3$ > PAN > NO. The flux budget analysis of Horii et al. (2005) in their initial presentation of this dataset supported the presence of an unidentified rapidly depositing $NO_y$ species in southwesterly flows, corroborating suggestions that alkyl nitrates resulting from oxidation of biogenic isoprene and monoterpenes in the presence of $NO_x$ could contribute up to 25 % of summertime $NO_y$ deposition at this site (Munger et al., 1998)."*

Finally, we remove a sentence from Section 4 describing an observed historical correlation of $NO_2$ uptake with RH to coniferous foliage:

(**removed from Section 4**) *"Thoene et al. (1996) monitored $NO_2$ uptake to Norway Spruce using a branch enclosure, proposing thin water films forming on needle surfaces as a plausible explanation for the observed correlation of $NO_2$ uptake to RH—a correlation which could not be explained by changes in stomatal conductance."*

This finding applied to daytime uptake at low RH (< 60%), which is tangential to the point of the preceding sentence (**L829**) that nocturnal growth of aqueous films at high RH (> 90%) could potentially occlude stomatal pores.

From Fig. 2 it looks like correcting for soil NO emission results in a relatively small increase in $V_{ex}$ (maybe 5 % to 10 % at most?). I suggest even simplifying Fig. 2 (bottom panel) by removing

the "observed" and just showing the soil NO corrected. In the caption a statement of "correcting for soil NO increased observed $V_{ex}$ by no more than 10 %" or something like that.

Response: Soil NO corrections estimated using a simulated canopy reduction factor (CRF) from parameterization P7 results in ~50% increase in observation-inferred $V_d(NO_2)$ averaged over the entire dataset (Table 4), and month-specific increases in the range 15% to 100% (Fig. 3). We would prefer to show both uncorrected and soil NO corrected $V_{ex}$ in Fig. 2 (bottom panel), so that the reduced influence of soil NO corrections on $V_{ex}$ at higher $NO_2$ concentrations can be seen. The revised manuscript now refers to both soil NO uncorrected and corrected $V_{ex}$ in the initial description of Fig. 2 (bottom panel):

**(changed in Section 3.3.1)** *"The bottom panel of Fig. 2 includes hourly values of $V_{ex}(NO_2)$ corrected for soil NO."* **changed to (L445)** *"The bottom panel of Fig. 2 includes hourly values of $V_{ex}(NO_2)$, both uncorrected and corrected for soil NO. "*

We also update the legend in Fig. 2 (bottom panel) to include more information on the soil NO correction:

**(change in Fig. 2 legend)** *"corrected for soil NO"* **changed to** *"corrected for soil NO with CRF following P7"*

and add a sentence following description of our soil NO correction methodology that explicitly states how the calculated canopy reduction factor varies with parameterization number in this study:

**(added to Section 3.3.1, L455)** *"As $NO_2$ surface uptake resistance $R_c(NO_2)$ is used in the calculation of CRF, values are parameterization specific (Fig. S3), with larger values of CRF resulting from lower values of $R_c(NO_2)$."*

In Section 3.3.3, while comparing to $NO_2$ surface resistances from other fields studies, we note that not considering soil NO corrections is reasonable under condition of high ambient $NO_2 > 10$ ppb. For Harvard Forest, it can be seen from Fig. 2 that soil NO has negligible influence on $V_{ex}(NO2)$ for NO2 concentrations above ~5 ppb. We think this is useful to point out, and add the following sentence to Section 3.3.3:

**(added to Section 3.3.3, L626)** *"The effect of soil NO on $V_{ex}(NO_2)$ as a function of above-canopy $NO_2$ concentration can be seen in Fig. 2, where negligible influence is noted for concentrations above ~ 5 ppb."*

Related to soil NO corrected and uncorrected depictions in Fig. 2., and in an effort to make Fig. 3 easier to understand and more transparent, we (i) depict standard deviations for observation-inferred $V_d(NO_2)$ uncorrected for soil NO rather than soil NO corrected following a specific parameterization and (ii) update the Number of Hourly Observations trace to depict the range in month-specific observation-inferred values resulting from differential outlier filter exceedance due to differences in simulated canopy reduction factors used in estimated soil NO corrections.

**(change in Fig. 3 caption)** *"Standard deviation about simulated monthly mean values, as well as measured monthly mean values corrected for soil NO using CRF from P7, are depicted as shaded areas"* **changed to (L472)** *"Standard deviations about simulated monthly mean values, as well as observation-inferred monthly mean values uncorrected for soil NO, are depicted as shaded areas. Month-specific range in number of hourly observations used in calculation of monthly means are indicated as vertical lines, and result from parameterization-specific soil NO corrections causing differential outlier filter exceedance (Fig. 2 and Table 4)."*

**Format changes in revised manuscript**

The following format changes have been made in the revised manuscript:

(1) *"trace gas"* changed to *"gaseous"* throughout when referring to gaseous dry deposition.

(2) *"employed"* changed to *"used"* throughout.

(3) *"observed"* changed to *"observation-inferred"* throughout when referring to eddy covariance $V_d(NO_2)$, in accordance with it being influenced through Eq. (6) (page 17, line 439).

(4) Parameterization P4 from the original manuscript has been removed from the revised manuscript, as discussed in response to RC2 above. Accordingly, parameterization numbers have been updated in the revised manuscript throughout, as listed in the updated Table 2.

**Correction of minor errors in revised manuscript**

We also made the following changes to correct minor errors in the revised manuscript. These changes do not affect our main conclusions.

(1) In supplemental Table S2, leaf surface area used to report foliar $V_d(NO_2)$ was misclassified as 'total leaf area' for references 8 & 9; this has been corrected to 'projected leaf area' and values have been updated accordingly. In addition, to improve clarity, we only include referenced values for foliar $V_d(NO_2)$ in Table S2 that contribute to the multi-study mean $NO_2$ uptake coefficients $\gamma$ reported in Table 1. Multi-study mean $\gamma$ for deciduous and coniferous leaves are reduced from $1.7 \times 10^{-6}$ (both) to $1.6 \times 10^{-6}$ (deciduous) and $1.5 \times 10^{-6}$ (coniferous) on a projected leaf area basis. Table 1 and S2 have been updated accordingly. Effect on bottom-up calculation of $V_d(NO_2)$ in Section 3.3.4 is less than 5% (Figure 4 and Table S3 updated accordingly) and doesn't change discussion of results; however, we update values in quantitative descriptions accordingly (less than 10% change). To acknowledge variability in leaf-level studies of $V_d(NO_2)$ from which we make a generalized prediction of nocturnal uptake between coniferous and deciduous species, and to encourage further evaluation in future studies, we add the following sentence to Section 4:

**(added to Section 4, L856)** *"However, given the intra- and interspecies variability in nocturnal $V_d^{surf}$ from leaf-level studies across a range of environmental conditions (i.e, Breuninger et al. (2013), Delaria et al. (2018, 2020), Rondón et al. (1993), and Wang et al. (2020)), further investigation into this simple generalization is warranted."*

(2) Table 1 and Section 2.4 previous had conflated the description of 'surface area scale factors' used for application of surface-specific $\gamma$ with 'surface area' used in surface-specific uptake experiments. While the intention of our descriptions is likely obvious to most readers, we have updated Table 1 and its description in Section 2.4 to avoid potential confusion:

**(added to Table 1, ~L280))** Column specifying surface area scale factors '$\alpha$ [unitless]' so as not to conflate with 'Surface Area'.

**(changed in Section 2.4)** (i)*"Care must be taken when comparing surface-specific $V_d^{surf}$ and $\gamma_{NO_2}$, as various surface area indices are used (i.e., planar, geometric, LAI, and total leaf area)."*

changed to **(L253)** *"care must be taken when comparing surface-specific $V_d^{surf}$ and $\gamma_{NO_2}$, as various surface areas are used (i.e., planar, geometric, projected leaf area, and total leaf area)."*

(ii) *"Others normalize to projected LAI as is routinely done..."* **changed to (L256)** *"Often, studies normalize to projected leaf area as is routinely done..."*

(iii) *"...surface area scale factors over which measured uptake was normalized..."* **changed to (L260)** *"...surface area scale factors α for which $\gamma_{NO_2}$ is to be applied..."*

Other locations throughout the manuscript in Section 3.3.4 and Section 4 have been updated to be consistent with these descriptions.

(3) In Section 3.4.1 when discussing a morning model-measurement discrepancy in $V_d(NO_y)$ in Fig. 6 following the diffusivity update in parameterization P4, Geddes et al. (2014) was incorrectly cited as a field study that found challenges in interpretation of morning $NO_y$ fluxes due to suspected canopy storage contributions. Geddes et al. (2014) noted an instance for a particular day where interpretation of $NO_y$ fluxes between 07:30–11:00 was confounded by possible canopy storage, and excluded a three-day period from campaign analysis, not all 07:30–11:00 periods as was claimed in Section 3.4.1. Accordingly, we have made the following correction in our discussion in Section 3.4.1:

**(changed in Section 3.4.1)** *"Geddes et al. (2014) measured eddy covariance fluxes of $NO_y$, NO, and $NO_2$ above two midlatitude mixed hardwood forests, noting challenges in interpretation of $NO_y$ fluxes between the hours of 07:30–11:00 LST due to suspected canopy storage contributions. Due to the absence of in-canopy $NO_y$ measurements from which to evaluate canopy storage, Geddes et al. (2014) considered observations of above-canopy $NO_y$ flux unrepresentative of the depositional flux over this timeframe and excluded observations during these hours from analysis. The anomalous morning peak in observation-inferred $V_d(NO_y)$ in Fig. 6 could, in part, be due to canopy storage contributions at Harvard Forest. It is conceivable that mixing down of rapidly depositing $NO_y$ species (i.e., $HNO_3$ or organic nitrates) as growth of the morning boundary layer erodes the residual layer above could induce a downward spike in above-canopy $NO_y$ flux. It is also possible that a morning low bias in simulated $V_d(NO_y)$ results from heavy reliance on the climatological diel profile of $HNO_3$ (Fig. 5) due to the paucity of hourly $HNO_3$ observations (Fig. 5 & Fig. S6)."* **changed to (L763)** *"Given the heavy reliance on*

*the climatological diel profile of HNO$_3$ (Fig. 5) due to the paucity of hourly HNO$_3$ observations (Fig. 5 & Fig. S6), and the potential influence of unmeasured rapidly depositing NO$_y$ species (Horii et al., 2005), further discussion of this model–measurement discrepancy is beyond the scope of this work; however, we note that it overlaps with the period of mixed layer growth."*

**(removed from Section 3.4.1)** *"... excluding the period from 07:00–11:00 reduces the daytime bias to -15 % and 24-hour bias to -17 %."*

This updated discussion is consistent with our scope of analysis and the existing concluding statement for Section 3.4.1: **(L804)** *"Horii et al. (2005) provide evidence of a rapidly depositing unidentified NO$_y$ species at this site, especially under southwesterly flow, and speculate the unidentified NO$_y$ species as organic nitrates."*

We have updated a sentence in the Conclusions to reflect this change:

**(changed in Conclusions)** *"Consequently, this improved representation of simulated V$_d$(NO$_y$) to an extensive dataset spanning many months wherein a potentially anomalous morning peak in observed V$_d$(NO$_y$) was exposed, in agreement with previous work demonstrating challenges in interpretation of late morning NO$_y$ fluxes over a similar forest ecosystem (Geddes and Murphy, 2014)."* **changed to (L866)** *"Consequently, this exposed a morning peak in observation-inferred V$_d$(NO$_y$), which simulated values failed to capture."*

(4) Corrected an error in a reported bias in a sentence in Section 3.4.1 beginning on **L767:** *"As seen for parameterization P8 in Fig. 6, updates to non-stomatal deposition of NO$_2$ and PAN result in noticeable increases in simulated V$_d$(NO$_y$) (16 % day; 30 % night), ..."*; should be *"8 % day"* not *"16 % day"*.

(5) In accordance with referenced data, model-measurement comparisons to observations from Talladega National Forest (Table 3) and Harvard Forest (Figs. 2–6 & Tables 4–5) were conducted using 'local time' and 'local standard time', respectively. However, we incorrectly stated 'local solar time' for both. We have corrected the time labels throughout figures, captions, and text of revised manuscript and supplement.

(6) Some units in the header of Table 3 were not written in exponential form. To conform to author guidelines, this has been corrected, as well as introducing abbreviations for $V_d$ and NMB in the Table's caption and $R_c$ and $z_o$ in the Table's sub notes.